# ProDAG: Projected Variational Inference for Directed Acyclic Graphs

**Ryan Thompson**[*]
University of Technology Sydney

**Edwin V. Bonilla**
CSIRO's Data61

**Robert Kohn**
University of New South Wales

## Abstract

Directed acyclic graph (DAG) learning is a central task in structure discovery and causal inference. Although the field has witnessed remarkable advances over the past few years, it remains statistically and computationally challenging to learn a single (point estimate) DAG from data, let alone provide uncertainty quantification. We address the difficult task of quantifying graph uncertainty by developing a Bayesian variational inference framework based on novel, provably valid distributions that have support directly on the space of sparse DAGs. These distributions, which we use to define our prior and variational posterior, are induced by a projection operation that maps an arbitrary continuous distribution onto the space of sparse weighted acyclic adjacency matrices. While this projection is combinatorial, it can be solved efficiently using recent continuous reformulations of acyclicity constraints. We empirically demonstrate that our method, `ProDAG`, can outperform state-of-the-art alternatives in both accuracy and uncertainty quantification.

## 1 Introduction

A directed acyclic graph (DAG) has directed edges between nodes and no directed cycles, allowing it to represent complex conditional independencies compactly. DAGs play a central role in causal inference (Spirtes et al., 2001; Pearl, 2009) and consequently have found applications spanning a broad range of domains such as psychology (Foster, 2010), economics (Imbens, 2020), and epidemiology (Tennant et al., 2021). The literature on DAGs, dating back at least to the 1980s (Lauritzen and Spiegelhalter, 1988; Pearl, 1988), has recently witnessed a resurgence of interest due to a computational breakthrough by Zheng et al. (2018) that formulated a continuous characterization for the discrete notion of acyclicity. This breakthrough opened the door for scalable first-order algorithms that can learn DAG structures from datasets containing many more variables than previously assumed feasible. We refer the reader to Vowels et al. (2022) and Kitson et al. (2023) for surveys of this work.

Unlike frequentist approaches that only learn point estimate DAGs (Zheng et al., 2018; Bello et al., 2022; Andrews et al., 2023), Bayesian approaches learn entire posterior distributions over DAGs. The posterior quantifies graph uncertainty while acknowledging that the true DAG might be identifiable only up to its Markov equivalence class.[2] Uncertainty quantification also benefits downstream tasks such as treatment effect estimation (Geffner et al., 2022) and experimental design (Annadani et al., 2023b). Most Bayesian approaches use a Gibbs-type prior based on a continuous acyclicity penalty (Annadani et al., 2021; Lorch et al., 2021; Geffner et al., 2022) or model DAGs in an augmented space where acyclicity is readily satisfied (Charpentier et al., 2022; Annadani et al., 2023a; Bonilla et al., 2024). While the former group (Gibbs-type priors) do not guarantee an exact posterior over DAGs, the latter group (augmented spaces) require relaxations or inference over discrete structures.

---

[*]Corresponding author. Email: `ryan.thompson-1@uts.edu.au`

[2]The Markov equivalence class is the set of DAGs that encode the same conditional independencies.

39th Conference on Neural Information Processing Systems (NeurIPS 2025).

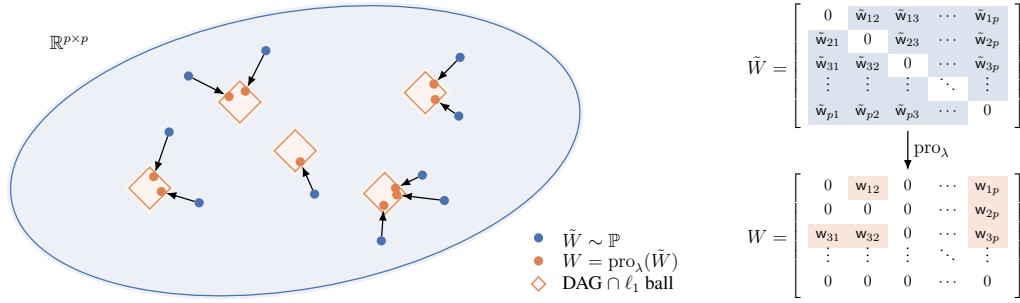

Figure 1: Illustration of `ProDAG`'s projected distributions. Samples (blue dots) from an unconstrained continuous space (blue ellipse) are projected onto the nearest acyclic matrix within an $\ell_1$-constrained region (orange diamonds). Projected samples (orange dots) satisfy acyclicity and sparsity constraints. Theoretically, we show that for any continuous $\tilde{W} \sim \mathbb{P}$ the projection $\text{pro}_\lambda(\tilde{W})$ is unique and measurable. This result implies that the projected distribution is a valid distribution over DAGs.

We propose `ProDAG`, a Bayesian method for learning DAGs that departs from the approaches above. Leveraging the power of variational inference, `ProDAG` models the prior and variational posterior using new distributions that natively have support on the space of DAGs. These DAG distributions are induced by mapping a continuous distribution $\mathbb{P}$ onto the space of DAGs via a minimal distance projection. Specifically, consider a DAG represented by the weighted adjacency matrix $W \in \mathbb{R}^{p \times p}$ with entry $\mathsf{w}_{jk}$ nonzero if and only if a directed edge exists from node $j$ to node $k$. We define a distribution over such matrices $W$, and hence over DAGs, via the data generating process

$$\tilde{W} \sim \mathbb{P}, \quad \lambda \sim \pi, \quad W = \text{pro}_\lambda(\tilde{W}), \tag{1}$$

where $\lambda \geq 0$ is a parameter governing the graph's sparsity (number of edges) with prior $\pi$ and

$$\text{pro}_\lambda(\tilde{W}) := \underset{W \in \text{DAG}, \, \|W\|_{\ell_1} \leq \lambda}{\arg\min} \frac{1}{2} \|\tilde{W} - W\|_F^2 \tag{2}$$

is the projection of the continuous matrix $\tilde{W} \in \mathbb{R}^{p \times p}$ onto the set of $\ell_1$ constrained acyclic matrices. Figure 1 visually illustrates the process. We prove this projection is a measurable mapping with a unique solution, yielding valid probability distributions. A key quality of these distributions is that they place probability mass on exact zeros, necessary for sampling valid, sparsely connected DAGs.

Inspired by recently developed continuous DAG characterizations (e.g., Zheng et al., 2018), we reformulate the combinatorial optimization problem (2) as an equivalent continuous optimization problem that can be solved in parallel on a GPU. We use this formulation in a variational framework by forming the prior and variational posterior using the new distributions, enabling scalable Bayesian DAG learning. Our framework accommodates DAGs represented as linear and nonlinear structural equation models (SEMs), making our approach suitable for problems and data of varying complexity. Experiments on synthetic and real data demonstrate that our approach often delivers more accurate inference than existing methods for DAG learning. Our toolkit for `ProDAG` is available on `GitHub`.

In summary, our **core contributions** are:

1. A new Bayesian model that directly samples from provably valid DAG distributions, ensuring exact acyclicity and eliminating the need for discrete approximations or relaxations;

2. A scalable variational inference framework that leverages GPU-accelerated projections, enabling efficient and accurate uncertainty quantification for linear and nonlinear DAGs;

3. An extensive suite of empirical evaluations that demonstrate state-of-the-art results across a wide variety of synthetic datasets, and validated on a real biological dataset; and

4. A user-friendly, well-documented, and open-source `Julia` implementation, made publicly available to promote adoption and reproducibility among researchers and practitioners.

## 2 Related work

Existing approaches for Bayesian inference on DAGs broadly fall into two groups: (1) regularization-based methods that include a continuous acyclicity penalty via a Gibbs-type prior in the variational objective (Annadani et al., 2021; Lorch et al., 2021; Geffner et al., 2022), and (2) state-augmentation methods that, e.g., place a prior over the graph's topological ordering (Viinikka et al., 2020; Cundy et al., 2021; Charpentier et al., 2022; Bonilla et al., 2024; Toth et al., 2024). The node-potential augmentation of Annadani et al. (2023a) is related to the latter category. See also Wang et al. (2022) and Deleu et al. (2023) for approaches based on probabilistic circuits and generative flow networks. Our approach differs by designing a prior and approximate posterior directly over DAGs without modeling the (discrete) topological ordering or using a penalty that does not place mass on zeros.

More broadly, our paper complements the literature on computational methods for point estimate DAG learning. The seminal paper of Zheng et al. (2018), which introduced the NOTEARS method, cleverly exploited the correspondence between the number of cycles in a matrix and the trace of that matrix's power. A similar connection is now known to hold for other continuous functions and is exploited by methods such as DAGMA (Bello et al., 2022), which we use in this work, as well as those of Ng et al. (2020), Yu et al. (2021), and Gillot and Parviainen (2022). These approaches that use continuous characterizations of acyclicity are appealing since they scale to large graphs and can be augmented with discrete combinatorics (Manzour et al., 2021; Deng et al., 2023) if required. Though not pursued here, our projection onto DAG space is also amenable to discrete techniques.

Finally, although distributions over DAGs via projections is a new idea, others have used projections to construct distributions over non-graphical structures. Xu and Duan (2023) designed distributions over sparse regression coefficients by projecting a vector drawn from an arbitrary continuous distribution onto the $\ell_1$ ball. They illustrated several computational advantages of their distributions as priors in a Hamiltonian Monte Carlo (HMC) framework over classical spike-and-slab priors and prove that the resulting posterior enjoys the minimax concentration rate. Xu et al. (2023) extended this work to general proximal mappings with HMC, which includes the convex $\ell_1$ projection as a special case. While our approach projects matrices onto the $\ell_1$ ball, we also impose acyclicity, giving rise to a nonconvex combinatorial optimization problem that does not constitute a proximal mapping.

## 3 Projected distributions

### 3.1 Description

To recap the introduction, we induce a distribution over weighted acyclic adjacency matrices $W$ as $W = \mathrm{pro}_\lambda(\tilde{W})$, where $\mathrm{pro}_\lambda$ is defined in (2), $\tilde{W} \sim \mathbb{P}$ is a continuous matrix, and $\lambda \sim \pi$ bounds the $\ell_1$ norm of $W$.[3] Since acyclic graphs have no self-loops (i.e., $W$ always has a zero diagonal), we can take $\mathbb{P}$ to be any continuous distribution over the off-diagonal elements of $\tilde{W}$ (e.g., a multivariate Gaussian), with the diagonal elements fixed as zero. The matrix $\tilde{W}$ can thus be interpreted as the weighted adjacency matrix of a *simple directed graph*, with the projection removing all cycles.

Besides removing cycles, the projection maps $\tilde{W}$ onto the $\ell_1$ ball of size $\lambda$. This part of the projection is necessary for sampling parsimonious graphs. Without this constraint, the projection would return the densest possible acyclic graph with $(p^2 - p)/2$ edges. The advantage of constraining $W$ to the $\ell_1$ ball, compared with modeling $\tilde{W}$ as a Laplace distribution, is that the $\ell_1$ ball gives probability mass on exact zeros beyond that already given by the acyclicity constraint. It is known that Laplace distributions do not concentrate exactly on zero except at their mode (Park and Casella, 2008).

On a conceptual level, the resulting distribution $p(W)$ can be understood as a mix of continuous components and exact zeros.[4] To elaborate, first observe that the nonzeros in $W$ may be indexed by a sparsity pattern $S \in \{0, 1\}^{p \times p}$ (i.e., an unweighted acyclic adjacency matrix). The nonzeros are then $W_S$, where $W_S$ is the restriction of $W$ to $S$. Because the $\ell_1$ constraint shrinks the nonzeros towards zero, the continuous $W_S$ is a transformation $f_\lambda$ of $\tilde{W}_S$, i.e., $W_S = f_\lambda(\tilde{W}_S)$, where $f_\lambda$ is implicit in

---

[3]The distribution $\pi$ can be a Dirac for a distribution over graphs of fixed sparsity (same number of edges) or an exponential for a distribution over graphs of varying sparsity.

[4]This mixture means that the projected distribution is multimodal, as different regions of the continuous distribution map to distinct acyclic graphs (except in degenerate point-mass cases).

the projection. This mixture interpretation suggests an interesting representation of $p(W)$:

$$p(W) \equiv \sum_{i=1}^{m} p(W_{S_i} \mid S_i)p(S_i) = \sum_{i=1}^{m} p(f_\lambda(\tilde{W}_{S_i}) \mid S_i)p(S_i),$$

where $S_1, \ldots, S_m$ is the finite collection of all acyclic adjacency matrices. We emphasize that this representation is purely conceptual and therefore we do not define or use the corresponding distributions over $S$, $W_S$, or $\tilde{W}_S$. However, it highlights that $p(W)$ forms a distribution over acyclic adjacency matrices $S$ without explicitly having to model the discrete, high-dimensional probability mass function $p(S)$. Moreover, as we later show, the projection can be reformulated as a continuous optimization problem, meaning one can sample from $p(W)$ without combinatorial complexity.

## 3.2 Properties

One might ask if $p(W)$ is a valid distribution. Theorem 1 addresses this question in the affirmative by showing that the projection is almost surely unique and measurable. The proof is in Appendix A.

**Theorem 1.** *Let $\tilde{W}$ be endowed with a continuous probability measure. Then it holds:*

1. *Projection (2) is unique almost surely with respect to the measure on $\tilde{W}$; and*

2. *Projection (2) is measurable with respect to the measure on $\tilde{W}$.*

Theorem 1 states that, for any continuous $\tilde{W} \sim \mathbb{P}$, the function $\mathrm{pro}_\lambda(\tilde{W})$ is uniquely defined and measurable. Uniqueness here means that for almost every $\tilde{W}$, the projection has a single minimizer; different values of $\tilde{W}$ may still map to the same $W$, but no single $\tilde{W}$ admits multiple optimal projections. This important result implies that a projected distribution is a valid distribution since a measurable function $g$ applied to a random variable $z$ produces a well-defined random variable $g(z)$.

# 4 Scalable projections

## 4.1 Continuous reformulation

To facilitate scalability, we reformulate the combinatorial projection (2) as an *equivalent* continuous optimization problem. This recharacterization follows recent work (Zheng et al., 2018; Bello et al., 2022) that established a useful and important equivalence relationship for certain functions $h(W)$:

$$h(W) = 0 \iff W \in \mathrm{DAG}.$$

Bello et al. (2022) showed that for all matrices $W \in \mathbb{W} := \{W \in \mathbb{R}^{p \times p} : \rho(W \circ W) < 1\}$, where $\rho$ is the spectral radius and $\circ$ is the Hadamard product, the function $h(W) := -\log \det(I - W \circ W)$ satisfies this property. This function, which constitutes the main innovation of DAGMA, attains the value zero if and only if there are no cycles in $W$. Thus, its level set at zero exactly corresponds to the entire set of DAGs. Though it is a nonconvex function, $h(W)$ is useful because it is both continuous and differentiable in $W$. We thus exploit this equivalence of constraints and rewrite projection (2) as

$$\mathrm{pro}_\lambda(\tilde{W}) = \underset{W \in \mathbb{W}, \, h(W) = 0, \, \|W\|_{\ell_1} \leq \lambda}{\arg\min} \frac{1}{2}\|\tilde{W} - W\|_F^2. \tag{3}$$

As we discuss next, it is possible to (i) solve this projection using first-order methods and (ii) evaluate its gradients analytically. These attributes pave the way for scalable, gradient-based posterior learning.

## 4.2 Algorithms

To solve projection (3), we employ a two-step procedure that produces a solution on the intersection of sets. Step 1 projects the matrix $\tilde{W}$ onto the set of acyclic matrices. We call the result $\hat{W}$. Step 2 projects $\hat{W}$ onto the $\ell_1$ ball, yielding the final output $W$ that remains acyclic but is further sparse.

For the acyclicity projection in Step 1, we use a path-following algorithm similar to Bello et al. (2022). This algorithm reformulates the problem by shifting the acyclicity constraint into the objective:

$$\min_{W \in \mathbb{W}} f_\mu(W; \tilde{W}) := \frac{\mu}{2}\|\tilde{W} - W\|_F^2 + h(W). \tag{4}$$

Here, $\mu \geq 0$ is taken along a sequence $\mu^{(1)} > \mu^{(2)} > \cdots > \mu^{(T)}$. The path-following algorithm solves (4) at $\mu = \mu^{(t+1)}$ by using the solution at $\mu = \mu^{(t)}$ as an initialization point (see Appendix B). The limiting solution as $\mu \to 0$ is guaranteed to satisfy the acyclicity constraint $h(W) = 0$ (Lemma 6, Bello et al., 2022). Crucially, the optimization problem (4) is amenable to gradient descent. The gradient of $f_\mu$ readily follows from the gradient of $h$, given by $\nabla h(W) = 2(I - W \circ W)^{-\top} \circ W$.

For the $\ell_1$ projection in Step 2, we employ a matrix version of the $\ell_1$ projection algorithm developed by Duchi et al. (2008). This non-iterative algorithm soft-thresholds the elements of the weighted adjacency matrix $\hat{W}$ that is produced by Step 1. This second step results in the final matrix $W$, a solution to the original projection (3). For completeness, we provide the algorithm in Appendix B.

### 4.3 Gradients

In many Bayesian learning frameworks, including our variational framework, it is necessary to compute gradients. In our case, this computation is complicated by the presence of the projection since we need its gradients with respect to $\tilde{W}$ and $\lambda$. Though it is technically possible to differentiate through our algorithms using automatic differentiation, it is practically infeasible due to the iterative nature of the path-following algorithm. As it turns out, it is possible to analytically evaluate the gradients via the implicit function theorem. This technique involves differentiating the Karush-Kuhn-Tucker (KKT) conditions of the projection that implicitly define $W$ as a function of $\tilde{W}$ and $\lambda$. Intuitively, the KKT conditions express how optimality of $W$ is preserved under small perturbations of $\tilde{W}$ or $\lambda$, and the implicit function theorem formalizes this dependence to yield local expressions for how $W$ varies. The gradients from this technique are in Appendix C alongside their derivation.

## 5 Posterior learning

### 5.1 Prior and variational posterior

Suppose we have data $X \in \mathbb{R}^{n \times p}$ with $n$ observations on $p$ variables $x \in \mathbb{R}^p$. Let $p(W)$ be a projected prior distribution over the DAG $W$. The conditional likelihood $p(X \mid W)$ is given by the linear SEM $x = W^\top x + \varepsilon$, where $\varepsilon \sim N(0, \Sigma)$. Our task is to infer the posterior distribution $p(W \mid X)$, which is analytically intractable. Though our distribution admits convenient algorithms, it remains taxing to learn the posterior via traditional approaches due to the high-dimensionality of the posterior space. We thus approximate the posterior using variational inference, leading to ProDAG.

Variational inference (e.g., Blei et al., 2017) optimizes the parameters $\theta$ of an approximate posterior $q_\theta(W)$ such that it is nearest to the true posterior $p(W \mid X)$ in Kullback-Leibler (KL) divergence. Like the prior, the approximate posterior is a projected distribution. However, as we discuss later, performing variational inference on $W$ alone is hard because the projection is non-invertible (multiple $\tilde{W}$ project to the same $W$). Hence, we focus on learning the joint posterior over $W$ and $\tilde{W}$. The joint posterior does not suffer from non-invertibility because for any $W$ the associated $\tilde{W}$ is also observed.

Let $p(\tilde{W}, W \mid X)$ be the true joint posterior and let $q_\theta(\tilde{W}, W)$ be the variational joint posterior. Since $W$ is a deterministic function of $\tilde{W}$, the variational joint posterior is the product of the variational marginal posterior $q_\theta(\tilde{W})$ and a Dirac delta function: $q_\theta(\tilde{W}, W) = q_\theta(\tilde{W})\delta(W - \text{pro}_\lambda(\tilde{W}))$. The marginal $q_\theta(\tilde{W})$ can be a continuous distribution such as a multivariate Gaussian. In that case, the parameters $\theta$ contain means and covariances. Choosing $q_\theta(\tilde{W})$ is the same as choosing $\mathbb{P}$ in (1). For simplicity and clarity of exposition, we treat the sparsity parameter $\lambda$ as fixed, i.e., $\pi$ in (1) is a Dirac.

### 5.2 Evidence lower bound

As in standard variational inference, we proceed by maximizing the evidence lower bound (ELBO):

$$\text{ELBO}(\theta) = \text{E}_{q_\theta(\tilde{W}, W)}[\log p(X \mid \tilde{W}, W)] - \text{KL}[q_\theta(\tilde{W}, W) \parallel p(\tilde{W}, W)]. \tag{5}$$

The first term on the right-hand side of (5) is the expectation of the log-likelihood under the variational posterior. Since the log-likelihood depends only on $\tilde{W}$ through $W$, this term simplifies as

$$\text{E}_{q_\theta(\tilde{W}, W)}[\log p(X \mid \tilde{W}, W)] = \text{E}_{q_\theta(W)}[\log p(X \mid W)].$$

To estimate this expectation, we sample DAGs from $q_\theta(W)$ and evaluate the log-likelihood on these samples. To sample from $q_\theta(W)$, we need only sample from $q_\theta(\tilde{W}, W)$ and discard $\tilde{W}$. The second term on the right-hand side of (5) is the KL divergence between the variational posterior $q_\theta(\tilde{W}, W)$ and the prior $p(\tilde{W}, W) = p(\tilde{W})\delta(W - \mathrm{pro}_\lambda(\tilde{W}))$. Since $W$ is a deterministic function of $\tilde{W}$, it is not too difficult to see that the KL divergence between the joint posterior and prior is equal to the KL divergence between the marginal posterior and prior on $\tilde{W}$. Hence, the second term also simplifies as

$$\mathrm{KL}[q_\theta(\tilde{W}, W) \parallel p(\tilde{W}, W)] = \mathrm{KL}[q_\theta(\tilde{W}) \parallel p(\tilde{W})].$$

Provided that $q_\theta(\tilde{W})$ and $p(\tilde{W})$ are chosen from distributional families with closed-form density functions such as multivariate Gaussian distributions, this term is amenable to analytical evaluation. The gradients of Gaussians and other (more complex) distributions for $\tilde{W}$ are readily accommodated in the ELBO using the well-known reparameterization trick (see, e.g., Kingma and Welling, 2014).

The challenge mentioned earlier of learning the marginal posterior of $W$ arises due to the KL divergence appearing in (5). If we instead focus on the marginal posterior, the divergence term in (5) is between $q_\theta(W)$ and $p(W)$. With some work, one can show that the divergence between these terms involves the divergence between the (unknown) prior and posterior conditional distributions of $\tilde{W}$ given $W$ (since $W$ is non-invertible in $\tilde{W}$). In any case, this issue is avoided in the joint posterior.

## 5.3 Algorithm

In light of the above remarks, we maximize the ELBO using the gradient descent-based Algorithm 1.

---

**Algorithm 1** ProDAG

---

**Input:** Initialization $\theta$, data $X \in \mathbb{R}^{n \times p}$, learning rate $\eta > 0$, no. of DAG samples $L \in \mathbb{N}$
**while** Not converged **do**
    Sample $\tilde{W}^{(l)} \sim q_\theta(\tilde{W})$ and set $W^{(l)} = \mathrm{pro}_\lambda(\tilde{W}^{(l)})$ for $l = 1, \ldots, L$
    Compute $\hat{\mathrm{ELBO}}(\theta) = 1/L \sum_{l=1}^{L} \log p(X \mid W^{(l)}) - \mathrm{KL}[q_\theta(\tilde{W}) \parallel p(\tilde{W})]$
    Update $\theta \leftarrow \theta + \eta \nabla_\theta \hat{\mathrm{ELBO}}(\theta)$
**end while**
**Output:** Optimized parameters $\theta$

---

The most taxing part of Algorithm 1 is the step that samples the $W$ matrices for the expected log-likelihood. Specifically, at each iteration, the algorithm projects a sample of $L$ matrices $\tilde{W}$ onto the constrained space. This projection, formulated with the continuous acyclicity constraint, involves the inversion of $p \times p$ matrices, which has cubic complexity. Fortunately, these can be done in parallel on a GPU using a batched CUDA implementation of the projection algorithms in Appendix B. Meanwhile, the gradients (required in the update step for the variational parameters $\theta$) have quadratic complexity in $p$ (see Appendix C). Hence, the overall complexity of training ProDAG is $O(p^3)$.

Though cubic complexity is not ideal, most state-of-the-art frequentist and Bayesian approaches to DAG learning also have cubic complexity (e.g., Lorch et al., 2021; Bello et al., 2022; Annadani et al., 2023a). Moreover, cubic complexity is the *worst case* complexity. We investigate the *expected case* by plotting the average training time over a range of $p$ in Figure 2. This figure also reports the average time to perform a parallel projection of 100 matrices, since this is the main source of complexity. Though certainly not scaling linearly, the times are reasonable for many applications, with training taking under a minute for $p = 20$ variables and slightly more than an hour for $p = 100$ variables.

## 6 Nonlinear DAGs

### 6.1 Structural equation model

The discussion so far assumes the SEM is linear. However, our approach accommodates nonlinear SEMs of the form $x = f(x) + \varepsilon$, where $f : \mathbb{R}^p \to \mathbb{R}^p$ is a nonlinear function with an acyclic Jacobian matrix. Acyclicity of the Jacobian matrix is necessary for a DAG because $\partial f_k(x)/\partial \mathsf{x}_j \neq 0$ indicates an edge exists from node $j$ to node $k$. A standard choice of the function $f$ is a neural network structured as $f(x) = (f_1(x), \ldots, f_p(x))^\top$, where $f_k(x)$ is a feedforward sub-network for node $k$.

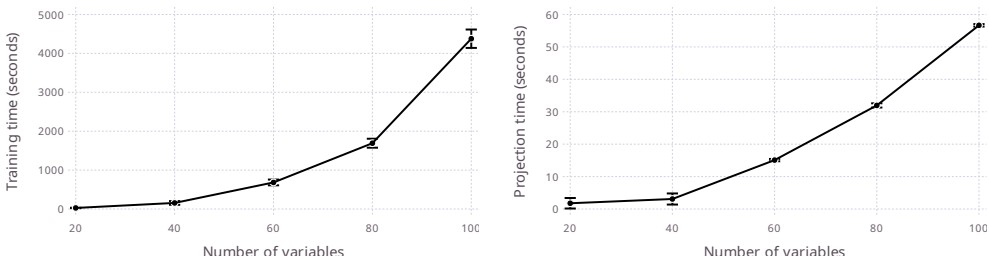

Figure 2: Computation times in seconds with a sample size $n = 100$. The averages (solid points) and standard errors (error bars) are measured over 10 independently and identically generated datasets.

Let $\omega = (\omega_1, \ldots, \omega_p)$ be a $p$-tuple of neural network weights corresponding to $f_1, \ldots, f_p$. Denote by $\omega_k^h \in \mathbb{R}^{p \times d}$ the restriction of the $k$th network's weights $\omega_k$ to the first hidden layer, where $d$ is the number of neurons in that layer. If the $j$th row of $\omega_k^h$, denoted $\omega_{jk}^h \in \mathbb{R}^d$, is a zero vector (i.e., $\|\omega_{jk}^h\|_2 = 0$), then $\partial f_k(x)/\partial \mathsf{x}_j = 0$ and no edge exists from node $j$ to node $k$. Hence, we can force the Jacobian of $f(x)$ to be acyclic by constraining the $p \times p$ matrix $W(\omega) := [\|\omega_{jk}^h\|_2]$ to be acyclic.

## 6.2 Posterior learning

Towards a nonlinear DAG, we construct a distribution over *all* weights $\omega$ of the neural network $f(x)$:
$$\tilde{\omega} \sim \mathbb{P}, \quad \lambda \sim \pi, \quad \omega = \mathrm{pro}_\lambda(\tilde{\omega}),$$
where $\mathbb{P}$ is a continuous distribution over the network weights. This data generating process yields a *fully Bayesian* neural network with every layer having a prior over its weights. The projection returns the weights $\omega$ nearest to $\tilde{\omega}$ such that the resulting adjacency matrix $W(\omega)$ is acyclic and sparse:
$$\mathrm{pro}_\lambda(\tilde{\omega}) := \underset{\substack{W(\omega) \in \mathbb{W}, \, h(W(\omega)) = 0 \\ \|W(\omega)\|_{\ell_1} \leq \lambda}}{\arg\min} \frac{1}{2} \sum_{k=1}^p \|\tilde{\omega}_k - \omega_k\|_F^2. \tag{6}$$

The matrix $W(\omega)$ forming the constraints in (6) is a function only of the *first* hidden layer weights. Hence, the projection only modifies those weights and leaves the remaining layer weights untouched.

Though the number of weights on the input layer is more numerous than the number of weights in the linear SEM case by a factor of $d$, the projection reduces to a simpler problem that is no more complex than the linear case. Proposition 1 presents this result. The proof is in Appendix D.

**Proposition 1.** *Let $\tilde{\omega}$ be a tuple of neural network weights and let $\tilde{\omega}_{jk}^h$ be the vector of weights for the $j$th input to the first hidden layer of the $k$th output. Define the matrix $\tilde{W}$ elementwise as $\tilde{\mathsf{w}}_{jk} = \|\tilde{\omega}_{jk}^h\|_2$ and let $W$ be the projection of $\tilde{W}$ in (3). Then projection (6) is solved by taking*
$$\omega_{jk}^h = \tilde{\omega}_{jk}^h \frac{\mathsf{w}_{jk}}{\tilde{\mathsf{w}}_{jk}}$$
*for $j, k = 1, \ldots, p$ and leaving the remaining layers' weights unmodified.*

Proposition 1 states that we need only project the matrix $\tilde{W}$ constructed from $\tilde{\omega}$ to obtain the weights $\omega$ for an acyclic neural network $f(x)$. Consequently, the nonlinear case requires no new algorithms.

Performing posterior inference on the neural network weights $\omega$ within our variational framework requires no major modifications. The only change required is to re-express the ELBO in terms of $\omega$:
$$\mathrm{ELBO}(\theta) = \mathrm{E}_{q_\theta(\omega)}[\log p(X \mid \omega)] - \mathrm{KL}[q_\theta(\tilde{\omega}) \,\|\, p(\tilde{\omega})],$$
where the variational parameters $\theta$ now parameterize a distribution of the same dimension as $\omega$.

## 7 Experiments

### 7.1 Baselines and metrics

Our method ProDAG is benchmarked against several state-of-the-art methods: DAGMA (Bello et al., 2022), DiBS and DiBS+ (Lorch et al., 2021), BayesDAG (Annadani et al., 2023a), and BOSS (Andrews

et al., 2023). `DAGMA` is a frequentist version of our approach that delivers a point estimate DAG. `DiBS`, `DiBS+`, and `BayesDAG` are alternative Bayesian approaches that learn posterior distributions over DAGs. `BOSS` is a frequentist comparator that relies on discrete (greedy search) methods rather than a continuous acyclicity characterization. Appendix E describes hyperparameters and implementations.

We consider several metrics that characterize different qualities of the learned posterior distributions. As a measure of uncertainty quantification, we report the Brier score under the posterior. As measures of structure recovery, we report the expected structural Hamming distance (SHD) and expected F1 score under the posterior. Finally, as a measure of discriminative power, we report the AUROC under the posterior. Lower values of Brier score and expected SHD are better, while higher values of expected F1 score and AUROC are better. All four metrics compare the learned posterior to the ground truth DAG. Since `DAGMA` and `BOSS` return a single point estimate graph, the metrics are evaluated by taking its posterior as a Dirac delta distribution that places all mass on the learned graph. For all other methods, the metrics are evaluated on a sample of graphs from the learned posterior.

## 7.2 Linear synthetic data

Figures 3 and 4 report results on datasets simulated from linear Erdős–Rényi DAGs with $p = 20$ and $p = 100$ nodes ($s = 40$ and $s = 200$ edges). The sample size varies from $n = 10$ to $n = 1000$.

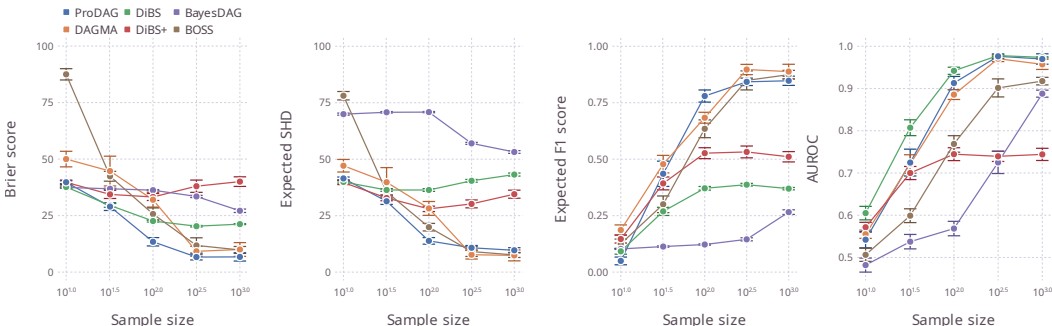

Figure 3: Performance on synthetic datasets generated from linear Erdős–Rényi DAGs with $p = 20$ nodes, $s = 40$ edges, and Gaussian noise. The averages (solid points) and standard errors (error bars) are measured over 10 independently and identically generated datasets.

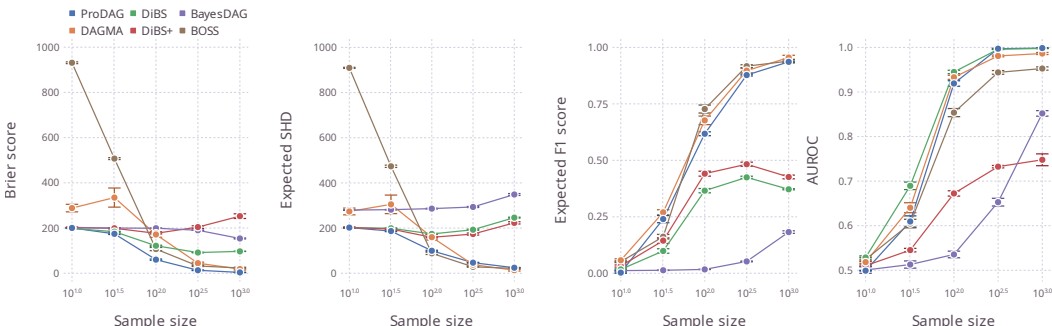

Figure 4: Performance on synthetic datasets generated from linear Erdős–Rényi DAGs with $p = 100$ nodes, $s = 200$ edges, and Gaussian noise. The averages (solid points) and standard errors (error bars) are measured over 10 independently and identically generated datasets.

Appendix F details the full setup. `ProDAG` delivers comparatively strong performance across the full suite of sample sizes, especially regarding uncertainty quantification. For small sample sizes, where there is little hope of recovering the true graph, `ProDAG` is as competitive as other Bayesian methods. Meanwhile, for moderate and large sample sizes, `ProDAG` overtakes the Bayesian methods across most metrics. `ProDAG` performs approximately as well as the frequentist `DAGMA` and `BOSS` for

the largest sample size, where little uncertainty remains regarding the true graph. Except for their discriminative power (AUROC), `DiBS` and `BayesDAG` do not significantly improve with the sample size. Lorch et al. (2021) studied `DiBS` on a sample size of $n = 100$, and their results roughly match ours at $n = 100$. Annadani et al. (2023a) evaluated `BayesDAG` on smaller graphs of size $p = 5$.

## 7.3 Nonlinear synthetic data

Figures 5 and 6 report results on datasets simulated from nonlinear DAGs. Following the setup in

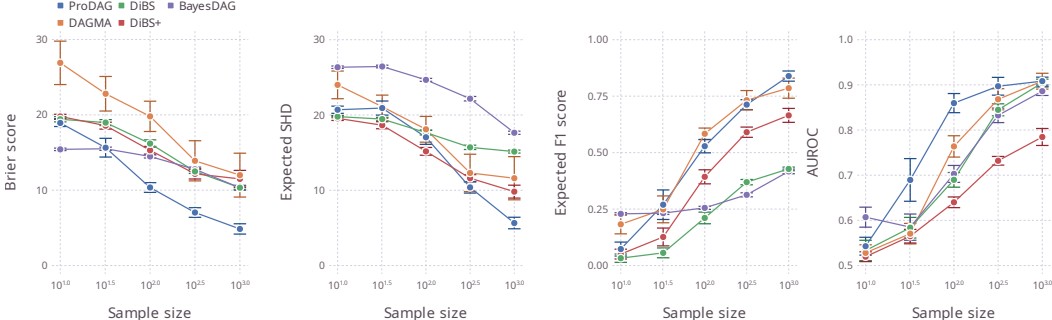

Figure 5: Performance on synthetic datasets generated from nonlinear Erdős–Rényi DAGs with $p = 10$ nodes, $s = 20$ edges, and Gaussian noise. The averages (solid points) and standard errors (error bars) are measured over 10 independently and identically generated datasets.

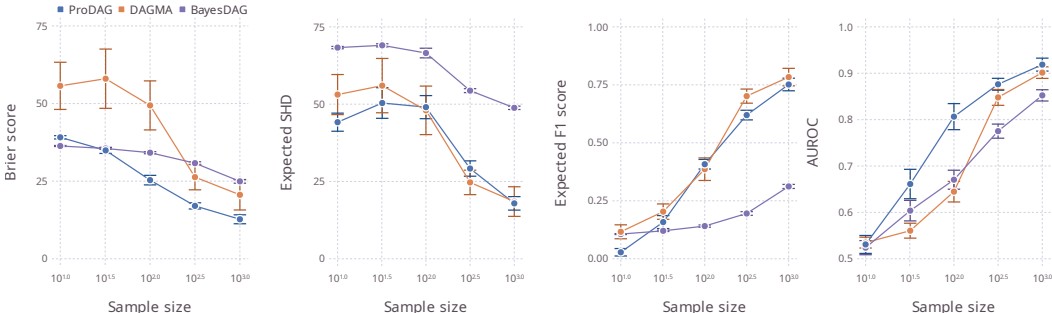

Figure 6: Performance on synthetic datasets generated from nonlinear Erdős–Rényi DAGs with $p = 20$ nodes, $s = 40$ edges, and Gaussian noise. The averages (solid points) and standard errors (error bars) are measured over 10 independently and identically generated datasets.

Zheng et al. (2020), each variable in the graph is generated from a single hidden-layer neural network whose inputs are that variable's parents. The learning task is more challenging than in the linear setting (more than 10× parameters in each graph), so we study graphs with $p = 10$ and $p = 20$ nodes ($s = 20$ and $s = 40$ edges). `BOSS` cannot model nonlinear DAGs, so it is excluded. `DiBS` and `DiBS+` exceed available memory when $p = 20$, so are excluded in that setting. `ProDAG` delivers strong performance and improves over the frequentist and Bayesian baselines most of the time. Again, it particularly excels at uncertainty quantification. The results for `DiBS` approximately match those in Lorch et al. (2021), where its expected SHD is similar to that of the null graph. Annadani et al. (2023a) studied `BayesDAG` on a sample size of $n = 5000$, much bigger than the largest sample size here. `ProDAG` does not require a sample of similarly large size to produce high-quality inference.

## 7.4 Real data

The flow cytometry data of Sachs et al. (2005) is a biological dataset designed to aid the discovery of protein signaling networks. The dataset contains $n = 7466$ human cell measurements on $p = 11$ phosphoproteins and phospholipids. To our knowledge, it is the only real dataset with an expert

consensus graph available, so it is widely used for real-world evaluation of DAG learning approaches. Table 1 reports results on the flow cytometry dataset, with the consensus graph consisting of 18 directed edges taken as the ground truth. `ProDAG` outperforms all approaches across all metrics. It

Table 1: Performance on the flow cytometry dataset of Sachs et al. (2005). The averages and standard errors are measured over 10 splits of the data. The best value of each metric is indicated in bold.

|  | Brier score | Exp. SHD | Exp. F1 score (%) | AUROC (%) |
|---|---|---|---|---|
| ProDAG | $16 \pm 0.2$ | $18 \pm 0.2$ | $24 \pm 0.6$ | $60 \pm 1.6$ |
| DAGMA | $36 \pm 2.1$ | $29 \pm 1.6$ | $17 \pm 1.4$ | $49 \pm 1.3$ |
| DiBS | $16 \pm 0.2$ | $22 \pm 0.1$ | $16 \pm 0.7$ | $55 \pm 1.6$ |
| DiBS+ | $26 \pm 1.1$ | $24 \pm 1.0$ | $15 \pm 2.7$ | $51 \pm 1.4$ |
| BayesDAG | $17 \pm 0.4$ | $25 \pm 0.5$ | $20 \pm 1.0$ | $59 \pm 1.6$ |
| BOSS | $34 \pm 0.7$ | $28 \pm 0.5$ | $15 \pm 1.3$ | $49 \pm 0.8$ |

achieves the lowest Brier score, tied with `DiBS`, and surpasses `DiBS+` and `BayesDAG` in uncertainty quantification. Additionally, `ProDAG` demonstrates superior performance in expected SHD, expected F1 score, and AUROC, reinforcing its strong overall capability in recovering the consensus graph structure. Meanwhile, `ProDAG` uniformly improves on `DAGMA`, highlighting the Bayesian benefits of `ProDAG` over `DAGMA`, since both are identical in their continuous characterization of acyclicity.

### 7.5   Additional experiments

A suite of additional experiments are included in Appendix G. In particular, Appendix G.1 presents results from further synthetic experiments covering (i) denser graphs, (ii) scale-free graphs, (iii) non-Gaussian noise distributions, and (iv) heteroscedastic noise distributions. Appendix G.2 reports results on semi-synthetic datasets generated from a large real-world medical diagnostic network. Appendix G.3 provides comparisons with a Markov chain Monte Carlo (MCMC) baseline. Finally, Appendix G.4 compares the run times of `ProDAG` with those of its primary Bayesian competitors.

## 8   Concluding remarks

DAGs are complex structures due to their acyclic nature, making Bayesian inference a formidable task. `ProDAG` is a novel approach that employs new projected distributions that have support directly on the space of DAGs. We study key properties of these distributions and show that they facilitate state-of-the-art Bayesian inference for linear and nonlinear DAGs in a scalable variational framework. While `ProDAG` is computationally costlier than frequentist methods, it is well-suited to settings where reliable posterior uncertainty is critical (e.g., decision making), where limited data make sparsity valuable (e.g., biology), or where strict acyclicity and interpretability are essential (e.g., policy).

As with other approaches to structure discovery using purely observational data, identifiability of the DAG structure in `ProDAG` depends on the correspondence between the assumed likelihood and the true data generating process. When the likelihood correctly specifies the noise distribution implied by the underlying SEM, standard identifiability results apply under assumptions such as causal sufficiency, the Causal Markov condition, and faithfulness. For instance, in a linear SEM with homoscedastic Gaussian noise, the DAG becomes fully identifiable (Peters and Bühlmann, 2014). In this setting, which our experiments cover, `ProDAG`'s posterior increasingly concentrates on the correct DAG as the sample size grows. Conversely, if the likelihood is misspecified, different graphs may induce indistinguishable likelihoods, leading to a loss of identifiability. In such cases, `ProDAG` represents the remaining structural ambiguity through the spread of its posterior distribution.

Recent work has extended continuous acyclicity approaches to interventional data (Brouillard et al., 2020; Faria et al., 2022; Lopez et al., 2022). Though our focus is observational data, `ProDAG` could be adapted for interventional settings by replacing the observational likelihood with one appropriate for interventions while leaving the projected distributions unchanged. Developing this extension and assessing `ProDAG` in interventional contexts offers a natural and important next step for research.

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

# A Proof of Theorem 1

*Proof.* We proceed by proving each claim of the theorem in turn.

**Uniqueness**   We begin by proving the first claim that projection (2) is unique almost surely. We assume hereafter that $\lambda > 0$ (if $\lambda = 0$ uniqueness holds trivially). First, observe that the weighted adjacency matrix $W$ can be expressed in terms of an unweighted (binary) adjacency matrix $S \in \{0,1\}^{p \times p}$ and a continuous matrix $V \in \mathbb{R}^{p \times p}$:

$$\min_{W \in \mathrm{DAG}, \|W\|_{\ell_1} \leq \lambda} \|\tilde{W} - W\|_F^2 \iff \min_{\substack{S \in \{0,1\}^{p \times p}, S \in \mathrm{DAG} \\ V \in \mathbb{R}^{p \times p}, \|V\|_{\ell_1} \leq \lambda}} f_S(V; \tilde{W}) := \|\tilde{W} - S \odot V\|_F^2,$$

where $\odot$ denotes the Hadamard product and we adopt the convention that $\mathsf{v}_{jk} = 0$ if $\mathsf{s}_{jk} = 0$. We can write the minimization on the right-hand side as a min-min problem:

$$\min_{\substack{S \in \{0,1\}^{p \times p}, S \in \mathrm{DAG} \\ V \in \mathbb{R}^{p \times p}, \|V\|_{\ell_1} \leq \lambda}} f_S(V; \tilde{W}) \iff \min_{\substack{S \in \{0,1\}^{p \times p} \\ S \in \mathrm{DAG}}} \min_{\substack{V \in \mathbb{R}^{p \times p} \\ \|V\|_{\ell_1} \leq \lambda}} f_S(V; \tilde{W}).$$

Consider the inner optimization problem on the right-hand side. For any $\tilde{W}$, let $V^\star(\tilde{W})$ denote an optimal solution to this problem, and define the set of $\tilde{W}$ with non-unique solutions as

$$A_S := \{\tilde{W} \in \mathbb{R}^{p \times p} \mid f_S(V_1^\star(\tilde{W}); \tilde{W}) = f_S(V_2^\star(\tilde{W}); \tilde{W}), \ V_1^\star(\tilde{W}) \neq V_2^\star(\tilde{W})\}.$$

Since $f_S(V; \tilde{W})$ is strictly convex in $V$ and the $\ell_1$ ball is convex, the optimal solution $V^\star(\tilde{W})$ is unique for all $\tilde{W}$. Hence, the set $A_S$ is empty. Now, define the set of $\tilde{W}$ with non-unique solutions across any two different adjacency matrices $S_1$ and $S_2$ as

$$A_{S_1, S_2} := \{\tilde{W} \in \mathbb{R}^{p \times p} \mid f_{S_1}(V_1^\star(\tilde{W}); \tilde{W}) = f_{S_2}(V_2^\star(\tilde{W}); \tilde{W}), \ V_1^\star(\tilde{W}) \neq V_2^\star(\tilde{W})\}.$$

Because $V_1^\star(\tilde{W}) \neq V_2^\star(\tilde{W})$, there is at least one coordinate $(j, k)$ on which they differ. Equality of the two quadratic objectives then forces $\tilde{W}$ to lie on a single affine hyperplane in $\mathbb{R}^{p \times p}$. Such a hyperplane has measure zero when $\tilde{W}$ is endowed with a continuous probability measure. Moreover, because there are finitely many pairs $(S_1, S_2)$ with $S_1 \neq S_2$, the union of $A_{S_1, S_2}$ over all pairs also has measure zero. We thus conclude that projection (2) is unique almost surely with respect to $\tilde{W}$.

**Measurability**   To prove the second claim that projection (2) is measurable, define the functions

$$g_\lambda(\tilde{W}, S) := \arg\min_{\substack{V \in \mathbb{R}^{p \times p} \\ \|V\|_{\ell_1} \leq \lambda}} \frac{1}{2} \|\tilde{W} - S \odot V\|_F^2,$$

where we again adopt the convention that $\mathsf{v}_{jk} = 0$ if $\mathsf{s}_{jk} = 0$, and

$$h_\lambda(\tilde{W}) := \arg\min_{\substack{S \in \{0,1\}^{p \times p} \\ S \in \mathrm{DAG}}} \frac{1}{2} \|\tilde{W} - S \odot g_\lambda(\tilde{W}, S)\|_F^2,$$

where, if the minimizer is not unique, we choose the lexicographically smallest $S$. It follows that

$$\mathrm{pro}_\lambda(\tilde{W}) = g_\lambda(\tilde{W}, h_\lambda(\tilde{W})).$$

We now demonstrate that each of these functions are measurable in $\tilde{W}$. First, for fixed $S$, observe that the function $g_\lambda(\tilde{W}, S)$ is the minimizing argument of a strictly convex optimization problem (with a compact feasible set) and $\tilde{W} \mapsto 1/2 \|\tilde{W} - S \odot V\|_F^2$ is continuous in $\tilde{W}$. Hence, by Berge's maximum theorem, $g_\lambda(\tilde{W}, S)$ is continuous in $\tilde{W}$. Continuous functions are measurable with respect to their argument, and therefore $g_\lambda(\tilde{W}, S)$ is measurable in $\tilde{W}$. Second, since the set of binary adjacency matrices is finite, the function $h_\lambda(\tilde{W})$ takes the pointwise minimum of finitely many measurable functions and returns the adjacency matrix indexed by that minimum (breaking any ties per the lexicographic order). A pointwise minimum of measurable functions is itself measurable. Lastly, since $g_\lambda(\tilde{W}, S)$ and $h_\lambda(\tilde{W})$ are both measurable in $\tilde{W}$, their composition $g_\lambda(\tilde{W}, h_\lambda(\tilde{W}))$ must also be measurable in $\tilde{W}$. We thus conclude that projection (2) is measurable with respect to $\tilde{W}$.

$\square$

# B Projection algorithms

Algorithm 2 provides the routine for the projection of the matrix $\tilde{W}$ onto the set of acyclic matrices. As discussed in the main text, the algorithm minimizes a penalized objective function along a

---

**Algorithm 2** Solver for acyclicity projection

---

**Input:** Continuous $\tilde{W} \in \mathbb{R}^{p \times p}$, initialization $W^{(0)} \in \mathbb{W}$, and scaling coefficients $\{\mu^{(t)}\}_{t=1}^{T}$

**for** $t = 0, 1, \ldots, T-1$ **do**

Initialize $W$ at $W^{(t)}$ and set

$$W^{(t+1)} \leftarrow \arg \min_{W \in \mathbb{W}} f_{\mu^{(t+1)}}(W; \tilde{W}) \tag{7}$$

**end for**

**Output:** Acyclic matrix $\hat{W} = W^{(T)}$

---

decreasing sequence of scaling coefficients $\mu^{(1)} > \cdots > \mu^{(T)}$. At each stage, the solution $W^{(t)}$ serves as a warm start for the next problem at $\mu^{(t+1)}$. This continuation scheme produces a sequence of unconstrained matrices $\{W^{(t)}\}_{t=1}^{T}$ that converges as $\mu \to 0$ to a point satisfying the constraint $h(W) = 0$. In our implementation, we set $\mu^{(1)} = 1$, $\mu^{(t+1)} = \mu^{(t)}/2$, $T = 10$, and $W^{(0)} = 0 \in \mathbb{W}$.

Algorithm 3 projects the acyclic matrix $\hat{W}$ from Algorithm 2 onto the $\ell_1$ ball of size $\lambda$. The algorithm

---

**Algorithm 3** Solver for $\ell_1$ projection

---

**Input:** Acyclic matrix $\hat{W} \in \mathbb{R}^{p \times p}$ and sparsity parameter $\lambda \geq 0$

Sort $v = \mathrm{vec}(\hat{W})$ as $|\mathsf{v}_1| \geq |\mathsf{v}_2| \geq \cdots \geq |\mathsf{v}_{p \times p}|$

Set

$$j_{\max} \leftarrow \max \left\{ j : |\mathsf{v}_j| > \left( \sum_{k=1}^{j} |\mathsf{v}_k| - \lambda \right) / j \right\}$$

$$\theta \leftarrow \sum_{j=1}^{j_{\max}} \left( |\mathsf{v}_j| - \lambda \right) / j_{\max}$$

Compute $W \in \mathbb{R}^{p \times p}$ elementwise as

$$\mathsf{w}_{jk} \leftarrow \mathrm{sign}(\hat{\mathsf{w}}_{jk}) \max(|\hat{\mathsf{w}}_{jk}| - \theta, 0)$$

**Output:** $\ell_1$ constrained acyclic matrix $W$

---

sorts the entries of $\hat{W}$ by absolute value, computes a threshold $\theta$ from the largest entries consistent with the $\ell_1$ constraint, and then applies soft-thresholding to shrink them toward zero (setting some exactly to zero) while preserving signs. The resulting matrix $W$ satisfies $\|W\|_{\ell_1} \leq \lambda$ by construction.

# C Gradients

Proposition 2 presents the gradients of the minimizer in (3) with respect to the continuous matrix $\tilde{W}$ and the sparsity parameter $\lambda$. The proof is at the end of this section.

**Proposition 2.** *Let* $W = (\mathsf{w}_{jk}) \in \mathbb{R}^{p \times p}$ *be the output of projection* (3) *applied to the matrix* $\tilde{W} = (\tilde{\mathsf{w}}_{jk}) \in \mathbb{R}^{p \times p}$ *with sparsity parameter* $\lambda > 0$*. Let* $\mathcal{A} := \{(j, k) : \mathsf{w}_{jk} \neq 0\}$ *be the set of active edge weights. Then, if the $\ell_1$ constraint is binding, the gradients of $\mathsf{w}_{jk}$ with respect to $\tilde{\mathsf{w}}_{qr}$ and $\lambda$ are*

$$\frac{\partial \mathsf{w}_{jk}}{\partial \tilde{\mathsf{w}}_{qr}} = \begin{cases} \delta_{jk}^{qr} - \dfrac{\mathrm{sign}(\mathsf{w}_{qr}) \, \mathrm{sign}(\mathsf{w}_{jk})}{|\mathcal{A}|} & \text{if } (j, k) \in \mathcal{A} \\ 0 & \text{otherwise} \end{cases}$$

*and*

$$\frac{\partial \mathsf{w}_{jk}}{\partial \lambda} = \begin{cases} \dfrac{\mathrm{sign}(\mathsf{w}_{jk})}{|\mathcal{A}|} & \textit{if } (j,k) \in \mathcal{A} \\ 0 & \textit{otherwise,} \end{cases}$$

*where $\delta_{jk}^{qr} := 1[(j,k) = (q,r)]$ and $|\mathcal{A}|$ is the cardinality of $\mathcal{A}$. Otherwise, if the $\ell_1$ constraint is non-binding, the gradients are*

$$\frac{\partial \mathsf{w}_{jk}}{\partial \tilde{\mathsf{w}}_{qr}} = \begin{cases} \delta_{jk}^{qr} & \textit{if } (j,k) \in \mathcal{A} \\ 0 & \textit{otherwise} \end{cases}$$

*and*

$$\frac{\partial \mathsf{w}_{jk}}{\partial \lambda} = 0.$$

Proposition 2 indicates that the Jacobian is sparse and has nonzeros only in positions corresponding to the surviving elements in $W$. In the binding case, where the $\ell_1$ constraint has a shrinkage effect, the value of the nonzeros depends on the size of the active set $\mathcal{A}$ (the number of nonzero edges). In the non-binding case, where the $\ell_1$ constraint imparts no shrinkage, all nonzero components equal one. Importantly, Proposition 2 indicates that only the projection's output $W$ is needed to evaluate the gradients; no additional computations are required. This result is useful for posterior learning since it means that a backward pass through the model demands virtually no further overhead cost.

Thompson et al. (2024) presented a result related to Proposition 2 in which they provided the gradients of a DAG projection for contextual learning. Their gradients differ slightly from those here because their sparsity constraint is different. The gradients relating to $\lambda$ were not considered in their work.

*Proof.* Let $W^\star$ denote an optimal solution to projection (3). The KKT conditions for stationarity and complementary slackness are

$$\partial \left( \frac{1}{2} \|\tilde{W} - W^\star\|_F^2 + \nu^\star(\|W^\star\|_{\ell_1} - \lambda) + \eta^\star h(W^\star) \right) \ni 0 \tag{8}$$

and

$$\nu^\star(\|W^\star\|_{\ell_1} - \lambda) = 0, \tag{9}$$

where $\nu^\star$ and $\eta^\star$ are the optimal values of the KKT multipliers. The optimal solution $W^\star$ is implicitly a function of the problem data $\tilde{W}$ and $\lambda$, i.e., $W^\star = W^\star(\tilde{W}, \lambda)$, as are the optimal KKT multipliers $\nu^\star = \nu^\star(\tilde{W}, \lambda)$ and $\eta^\star = \eta^\star(\tilde{W}, \lambda)$. Denote the set of nonzeros by $\mathcal{A}$.

**Non-binding $\ell_1$ constraint** Suppose the $\ell_1$ constraint is non-binding. Evaluating the subderivative on the left-hand side of the stationarity condition (8) gives

$$\mathsf{w}_{jk}^\star - \tilde{\mathsf{w}}_{jk} = 0,$$

where we use that $\nu^\star = 0$ and $\partial h(W^\star)/\partial \mathsf{w}_{jk}^\star = 0$. Differentiating this expression with respect to $\tilde{\mathsf{w}}_{qr}$ for $(j,k) \in \mathcal{A}$, gives

$$\frac{\partial \mathsf{w}_{jk}^\star}{\partial \tilde{\mathsf{w}}_{qr}} = \delta_{jk}^{qr}.$$

For those $(j,k) \notin \mathcal{A}$, $\mathsf{w}_{jk}^\star = 0$ and hence

$$\frac{\partial \mathsf{w}_{jk}^\star}{\partial \tilde{\mathsf{w}}_{qr}} = 0.$$

Moreover, because $\nu^\star = 0$, we have

$$\frac{\partial \mathsf{w}_{jk}^\star}{\partial \lambda} = 0$$

for all $(j,k)$.

**Binding $\ell_1$ constraint: active set**   Suppose the $\ell_1$ constraint is binding and consider the gradients for $\mathsf{w}_{jk}^\star$ in the active set. For these $(j, k) \in \mathcal{A}$, it follows from the stationarity condition (8) that

$$\mathsf{w}_{jk}^\star - \tilde{\mathsf{w}}_{jk} + \nu^\star \operatorname{sign}(\mathsf{w}_{jk}^\star) = 0, \tag{10}$$

where we again use that $\partial h(W^\star)/\partial \mathsf{w}_{jk}^\star = 0$. The complementary slackness condition (9) for $(j, k) \in \mathcal{A}$ gives

$$\sum_{(j,k)\in\mathcal{A}} |\mathsf{w}_{jk}^\star| - \lambda = 0. \tag{11}$$

We proceed by first deriving the gradients of $\mathsf{w}_{jk}^\star$ with respect to $\tilde{\mathsf{w}}_{qr}$. Differentiating (10) with respect to $\tilde{\mathsf{w}}_{qr}$ gives

$$\frac{\partial \mathsf{w}_{jk}^\star}{\partial \tilde{\mathsf{w}}_{qr}} - \delta_{jk}^{qr} + \frac{\partial \nu^\star}{\partial \tilde{\mathsf{w}}_{qr}} \operatorname{sign}(\mathsf{w}_{jk}^\star) + \nu^\star \frac{\partial}{\partial \tilde{\mathsf{w}}_{qr}} \operatorname{sign}(\mathsf{w}_{jk}^\star) = 0,$$

from which it follows

$$\frac{\partial \mathsf{w}_{jk}^\star}{\partial \tilde{\mathsf{w}}_{qr}} = \delta_{jk}^{qr} - \frac{\partial \nu^\star}{\partial \tilde{\mathsf{w}}_{qr}} \operatorname{sign}(\mathsf{w}_{jk}^\star). \tag{12}$$

Next, differentiating (11) with respect to $\tilde{\mathsf{w}}_{qr}$ gives

$$\sum_{(j,k)\in\mathcal{A}} \operatorname{sign}(\mathsf{w}_{jk}^\star) \frac{\partial \mathsf{w}_{jk}^\star}{\partial \tilde{\mathsf{w}}_{qr}} = 0. \tag{13}$$

Substituting (12) into (13) and solving for $\partial \nu^\star / \partial \tilde{\mathsf{w}}_{qr}$ yields

$$\frac{\partial \nu^\star}{\partial \tilde{\mathsf{w}}_{qr}} = \frac{\operatorname{sign}(\mathsf{w}_{qr}^\star)}{\sum_{(j,k)\in\mathcal{A}} \operatorname{sign}(\mathsf{w}_{jk}^\star)^2} = \frac{1}{|\mathcal{A}|} \operatorname{sign}(\mathsf{w}_{qr}^\star). \tag{14}$$

Finally, substituting (14) back into (12) gives

$$\frac{\partial \mathsf{w}_{jk}^\star}{\partial \tilde{\mathsf{w}}_{qr}} = \delta_{jk}^{qr} - \frac{\operatorname{sign}(\mathsf{w}_{qr}^\star) \operatorname{sign}(\mathsf{w}_{jk}^\star)}{|\mathcal{A}|}.$$

We turn now to the gradients of $\mathsf{w}_{jk}^\star$ with respect to $\lambda$. Differentiating (10) with respect to $\lambda$ gives

$$\frac{\partial \mathsf{w}_{jk}^\star}{\partial \lambda} + \frac{\partial \nu^\star}{\partial \lambda} \operatorname{sign}(\mathsf{w}_{jk}^\star) + \nu^\star \frac{\partial}{\partial \lambda} \operatorname{sign}(\mathsf{w}_{jk}^\star) = 0,$$

from which it follows

$$\frac{\partial \mathsf{w}_{jk}^\star}{\partial \lambda} = -\frac{\partial \nu^\star}{\partial \lambda} \operatorname{sign}(\mathsf{w}_{jk}^\star). \tag{15}$$

Next, differentiating (11) with respect to $\lambda$ gives

$$\sum_{(j,k)\in\mathcal{A}} \operatorname{sign}(\mathsf{w}_{jk}^\star) \frac{\partial \mathsf{w}_{jk}^\star}{\partial \lambda} - 1 = 0. \tag{16}$$

Substituting (15) into (16) and solving for $\partial \nu^\star / \partial \lambda$ yields

$$\frac{\partial \nu^\star}{\partial \lambda} = -\frac{1}{|\mathcal{A}|}. \tag{17}$$

Finally, substituting (17) back into (15) gives

$$\frac{\partial \mathsf{w}_{jk}^\star}{\partial \lambda} = \frac{\operatorname{sign}(\mathsf{w}_{jk}^\star)}{|\mathcal{A}|}.$$

**Binding $\ell_1$ constraint: inactive set**   Suppose the $\ell_1$ constraint is binding and consider the gradients for $\mathsf{w}_{jk}^\star$ in the inactive set. These $(j, k) \notin \mathcal{A}$ play no role in either of the KKT conditions, and hence have no gradient with respect to $\tilde{\mathsf{w}}_{jk}$ or $\lambda$:

$$\frac{\partial \mathsf{w}_{jk}^\star}{\partial \tilde{\mathsf{w}}_{qr}} = 0$$

and

$$\frac{\partial \mathsf{w}_{jk}^\star}{\partial \lambda} = 0.$$

$\square$

# D  Proof of Proposition 1

*Proof.* The acyclicity constraint forces $\|\omega_{jk}^h\|_2 = 0$ for some $j$ and $k$, meaning the whole vector $\omega_{jk}^h = 0$. Likewise, the sparsity constraint, which represents a group lasso regularizer (Yuan and Lin, 2006), is also known to set some $\omega_{jk}^h$ to zero. Those surviving vectors that are not set to zero are shrunk towards zero by a factor $\|\omega_{jk}^h\|_2/\|\tilde{\omega}_{jk}^h\|_2$. The combined effect of these two constraints gives

$$\omega_{jk}^h = \tilde{\omega}_{jk}^h \frac{\|\omega_{jk}^h\|_2}{\|\tilde{\omega}_{jk}^h\|_2},$$

which equals zero if $\|\omega_{jk}^h\|_2 = 0$. Now, observe that the objective function can be expressed solely in terms of the norms of each vector:

$$
\begin{aligned}
\sum_{j,k} \|\tilde{\omega}_{jk}^h - \omega_{jk}^h\|_2^2 &= \sum_{j,k} \left\| \tilde{\omega}_{jk}^h - \tilde{\omega}_{jk}^h \frac{\|\omega_{jk}^h\|_2}{\|\tilde{\omega}_{jk}^h\|_2} \right\|_2^2 \\
&= \sum_{j,k} \left\| \tilde{\omega}_{jk}^h \left( 1 - \frac{\|\omega_{jk}^h\|_2}{\|\tilde{\omega}_{jk}^h\|_2} \right) \right\|_2^2 \\
&= \sum_{j,k} \|\tilde{\omega}_{jk}^h\|_2^2 \left( 1 - \frac{\|\omega_{jk}^h\|_2}{\|\tilde{\omega}_{jk}^h\|_2} \right)^2 \\
&= \sum_{j,k} (\|\tilde{\omega}_{jk}^h\|_2 - \|\omega_{jk}^h\|_2)^2.
\end{aligned}
$$

Taking $\tilde{\mathsf{w}}_{jk} = \|\tilde{\omega}_{jk}^h\|_2$ and $\mathsf{w}_{jk} = \|\omega_{jk}^h\|_2$ yields the claim of the proposition. $\qquad\square$

# E  Implementation

`ProDAG` uses a multivariate Gaussian prior on $\tilde{W}$ with independent elements, each having mean zero and variance one. The variational posterior on $\tilde{W}$ is also taken as a multivariate Gaussian with independent elements, where the means and variances are initialized at their prior values. We use the Adam optimizer (Kingma and Ba, 2015) with a learning rate of 0.1 to minimize the variational objective function. The posterior variances are held positive during optimization using a softplus transform. At each Adam iteration, 100 samples of $\tilde{W}$ are drawn from the posterior to estimate the objective function. To project these sampled $\tilde{W}$ and obtain $W$, we solve the projection using gradient descent with learning rates of $1/p$ and $0.25/p$ for the linear and nonlinear settings, respectively. Similar to `DAGMA`, the resulting $W$ can contain small but nonzero elements that violate acyclicity. These small nonzeros occur because gradient descent seldom produces machine-precision zeros. To that end, we follow standard practice and set small elements of $W$ to zero with a threshold of 0.1. At inference time, this threshold can be set such that $W$ is guaranteed acyclic (see, e.g., Ng et al., 2020).

Our implementation of `ProDAG` uses the machine learning library `Flux` (Innes et al., 2018) and performs projections using parallel GPU (batched CUDA) implementations of the projection algorithms.

For the benchmark methods, we use the respective authors' open-source `Python` implementations:

- `DAGMA`: https://github.com/kevinsbello/dagma, v1.1.0, Apache License 2.0;
- `DiBS` and `DiBS+`: https://github.com/larslorch/dibs, v1.3.3, MIT License;
- `BayesDAG`: https://github.com/microsoft/Project-BayesDAG, v0.1.0, MIT License; and
- `BOSS`: https://github.com/cmu-phil/tetrad, v7.6.6, GNU General Public License v2.0.

We use 50 particles for `DiBS`'s particle variational inference. More particles exceed available memory.

The sparsity parameter $\lambda$ of each method is tuned over a grid of 10 values between $\lambda^{\min}$ and $\lambda^{\max}$:[5]

- `ProDAG`: $\lambda^{\min} = 0$ and $\lambda^{\max}$ is the average $\ell_1$ ball from the learned posterior with $\lambda = \infty$;
- `DAGMA`: $\lambda^{\min} = 10^{-3}$ and $\lambda^{\max} = 1$;
- `DiBS` and `DiBS+`: do not include a sparsity parameter;
- `BayesDAG`: $\lambda^{\min} = 10$ and $\lambda^{\max} = 10^3$; and
- `BOSS`: does not include a sparsity parameter.

The above grids are chosen so that the sparsity level matches the true sparsity level for some value between $\lambda^{\min}$ and $\lambda^{\max}$. We choose the specific value of $\lambda$ using a separate validation set of size $\lfloor 0.1n \rfloor$. We cannot extract the weighted adjacency matrices or neural networks from `BayesDAG` (only the binary adjacency matrices), so we choose its $\lambda$ so that it is closest to the true sparsity level.

In the nonlinear setting, the neural networks for `ProDAG` and `DiBS` consist of a single hidden layer with 10 neurons that use ReLU activation functions. `DAGMA` employs the same architecture but the neurons use sigmoid activations as its implementation does not support custom activation functions. The network for `BayesDAG` consists of two hidden layers, each with 128 neurons that use ReLU activations. `BayesDAG`'s implementation does not support custom numbers of layers or neurons.

The experiments are run on a Linux workstation with an AMD Ryzen Threadripper PRO 5995WX CPU, 256GB RAM, and $2 \times$ NVIDIA GeForce RTX 4090 GPUs. Each method is allocated either a single GPU or a single CPU core, with the experiments run in parallel across GPUs/CPU cores.

## F   Simulation design

The synthetic datasets generated from linear DAGs are constructed as follows. We first randomly sample an Erdős–Rényi or scale-free graph with $p$ nodes and $s$ edges. The edges of this undirected graph are then oriented according to a randomly generated topological ordering. Each directed edge in the graph is then weighted by sampling weights uniformly on $[-0.7, -0.3] \cup [0.3, 0.7]$, yielding a weighted adjacency matrix $W$. We then draw iid noise $\varepsilon_1, \ldots, \varepsilon_n$ and generate the variables as

$$x_i = W^\top x_i + \varepsilon_i, \quad i = 1, \ldots, n.$$

For synthetic datasets generated from nonlinear DAGs, we likewise sample an Erdős–Rényi or scale-free graph and orient it according to a random topological ordering. We then generate a feedforward neural network $f_k$ with a single hidden layer of 10 neurons for $k = 1, \ldots, p$. The weights of $f_k$ are sampled uniformly from $[-0.7, -0.3] \cup [0.3, 0.7]$, with hidden layer weights of node $k$'s non-parents set to zero. The activation is a ReLU. The noise $\varepsilon_1, \ldots, \varepsilon_n$ is drawn iid and the variables are taken as

$$x_i = f(x_i) + \varepsilon_i, \quad i = 1, \ldots, n,$$

where $f(x) = (f_1(x), \ldots, f_p(x))$.

The noise distribution is specified as follows. For the Gaussian experiments, we draw $p$ independent components from a $\mathrm{Normal}(0, 1)$ distribution. For the exponential experiments, the components are drawn from an $\mathrm{Exponential}(1)$ distribution, and for the Gumbel experiments from a $\mathrm{Gumbel}(0, 1)$ distribution. For heteroscedastic Gaussian experiments, we draw $\varepsilon_{ij}$ (the $j$th component of $\varepsilon_i$) from $\mathrm{Normal}(0, \sigma_j^2)$, where $\sigma_j$ is sampled from $\mathrm{Uniform}(2/3, 4/3)$ and held fixed across observations.

The Brier score, used as a metric in the experiments, is defined as

$$\mathrm{Brier} := \sum_{j,k} \left( 1[\mathsf{w}_{jk} \neq 0] - \hat{p}_{jk} \right)^2,$$

where $1[\mathsf{w}_{jk} \neq 0]$ is the indicator that the true DAG contains a directed edge from node $j$ to node $k$ and $\hat{p}_{jk}$ is the posterior probability that this edge exists, as estimated from posterior samples.

---

[5]The parameter $\lambda$ in `ProDAG` refers to the $\ell_1$ ball size, whereas $\lambda$ in `DAGMA` and `BayesDAG` refers to the coefficient on the $\ell_1$ penalty.

# G  Additional experiments

## G.1  Alternative simulation designs

We investigate the robustness of `ProDAG` to several alternative simulation designs. Figures 7 and 8 report results on dense graphs containing two and three times as many edges as nodes. Figures 9 and 10 report results on scale-free graphs whose degree exponent is set to two and three. Figures 11 and 12 report results from Gumbel noise distributions, while Figures 13 and 14 report results from exponential noise distributions. Finally, Figure 15 report results from Gaussian noise distributions with heteroscedastic variance. A Gaussian likelihood is used consistently across settings. Under all simulation designs, the relative performance of `ProDAG` is roughly similar to that in the main experiments, indicating that `ProDAG` remains reliable across a broad range of realistic scenarios.

## G.2  Semi-synthetic datasets

We generate semi-synthetic datasets using the MUNIN (subnetwork #1) DAG, a large medical diagnostic network from the Bayesian Network Repository (https://www.bnlearn.com/bnrepository). The graph, which contains $p = 186$ nodes and $s = 273$ edges, is used to generate semi-synthetic datasets by sampling edge weights and noise in the same fashion as the fully synthetic experiments. Table 2 reports results with sample size $n = 100$, showing that `ProDAG` performs strongly with the lowest Brier score and expected SHD. While `DAGMA` scores well on expected F1 score and AUROC,

Table 2: Performance on semi-synthetic datasets of size $n = 100$ generated from the MUNIN graph from the Bayesian Network Repository with $p = 186$ nodes, $s = 273$ edges, and Gaussian noise. The averages and standard errors are measured over 10 independently and identically generated datasets. The best value of each metric is indicated in bold.

|          | Brier score      | Exp. SHD        | Exp. F1 score (%) | AUROC (%)       |
| -------- | ---------------- | --------------- | ----------------- | --------------- |
| ProDAG   | **146 ± 4.0**    | **201 ± 4.9**   | 37 ± 2.0          | 87 ± 0.6        |
| DAGMA    | 305 ± 65.8       | 274 ± 64.9      | **67 ± 4.1**      | **89 ± 0.5**    |
| DiBS     | -                | -               | -                 | -               |
| DiBS+    | -                | -               | -                 | -               |
| BayesDAG | 279 ± 0.3        | 557 ± 0.9       | 1 ± 0.0           | 53 ± 0.7        |
| BOSS     | -                | -               | -                 | -               |

`ProDAG` provides superior calibration and structure recovery. To our knowledge, this experiment represents one of the largest Bayesian DAG evaluations reported to date. Aside from `ProDAG`, the only methods to successfully run are `DAGMA` and `BayesDAG`. The other baselines encounter difficulties: `DiBS` and `DiBS+` produce nonsensical results indicative of convergence issues, while `BOSS` crashes.

## G.3  MCMC baseline

We compare `ProDAG` with `Gadget` (Viinikka et al., 2020), an earlier Bayesian approach that learns posteriors over DAGs via MCMC. We focus on linear Erdős–Rényi DAGs with $p = 20$ nodes and $s = 40$ edges, as `Gadget` does not scale to our larger setting. Table 3 reports results with sample size $n = 100$, showing that `Gadget` substantially underperforms `ProDAG`. This underperformance is not

Table 3: Performance on synthetic datasets of size $n = 100$ generated from linear Erdős–Rényi DAGs with $p = 20$ nodes, $s = 40$ edges, and Gaussian noise. The averages and standard errors are measured over 10 independently and identically generated datasets. The best value of each metric is indicated in bold.

|        | Brier score   | Exp. SHD     | Exp. F1 score (%) | AUROC (%)     |
| ------ | ------------- | ------------ | ----------------- | ------------- |
| ProDAG | **13 ± 1.8**  | **14 ± 1.5** | **78 ± 2.7**      | **91 ± 1.6**  |
| Gadget | 52 ± 1.8      | 37 ± 0.5     | 17 ± 1.5          | 65 ± 3.0      |

due to `Gadget`'s configuration, as we evaluated many settings and report the best-performing one.

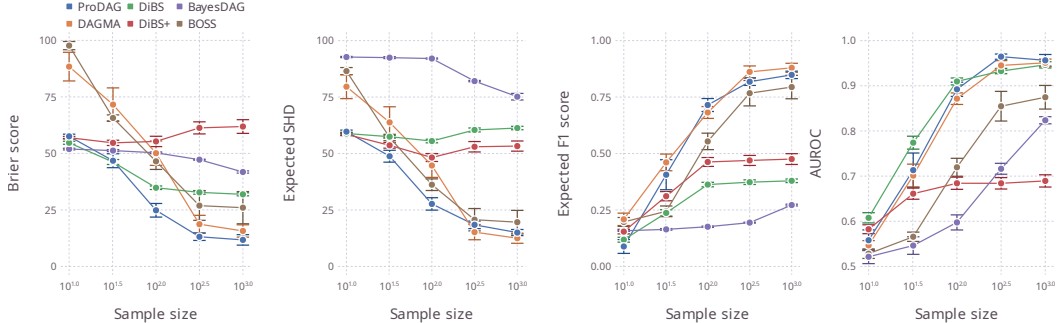

Figure 7: Performance on synthetic datasets generated from linear Erdős–Rényi DAGs with $p = 20$ nodes, $s = 60$ edges, and Gaussian noise. The averages (solid points) and standard errors (error bars) are measured over 10 independently and identically generated datasets.

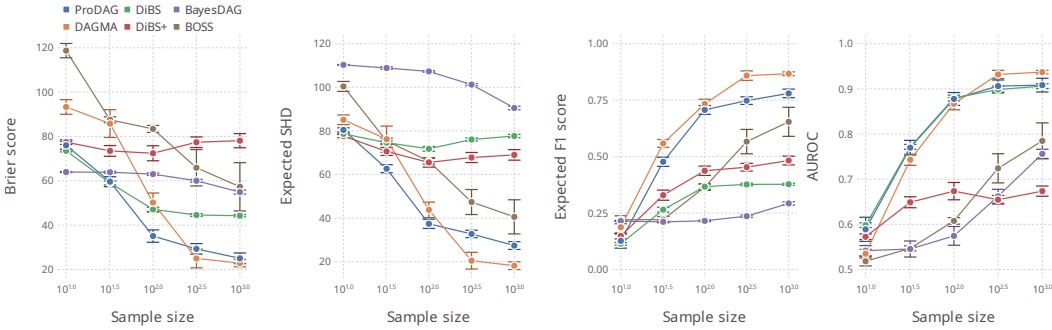

Figure 8: Performance on synthetic datasets generated from linear Erdős–Rényi DAGs with $p = 20$ nodes, $s = 80$ edges, and Gaussian noise. The averages (solid points) and standard errors (error bars) are measured over 10 independently and identically generated datasets.

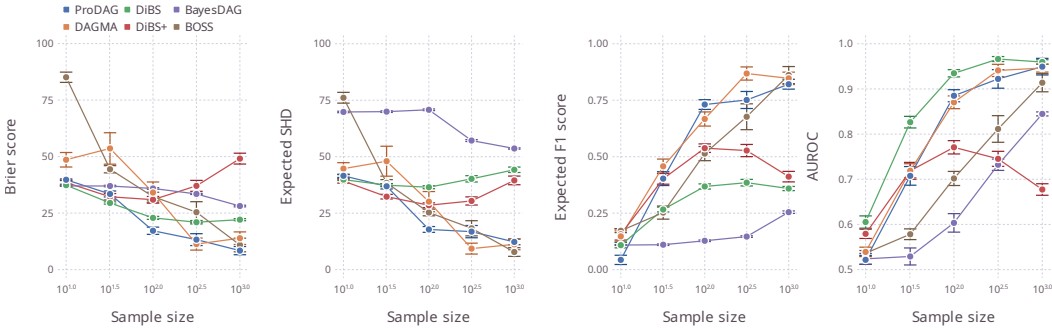

Figure 9: Performance on synthetic datasets generated from linear scale-free (degree exponent 2) DAGs with $p = 20$ nodes, $s = 40$ edges, and Gaussian noise. The averages (solid points) and standard errors (error bars) are measured over 10 independently and identically generated datasets.

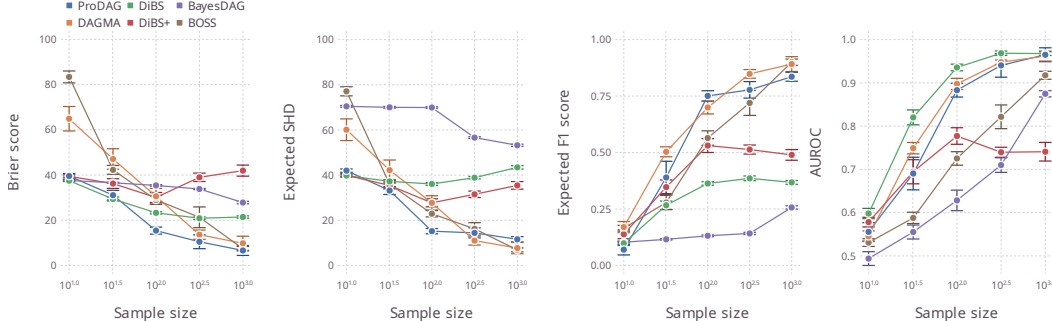

Figure 10: Performance on synthetic datasets generated from linear scale-free (degree exponent 3) DAGs with $p = 20$ nodes, $s = 40$ edges, and Gaussian noise. The averages (solid points) and standard errors (error bars) are measured over 10 independently and identically generated datasets.

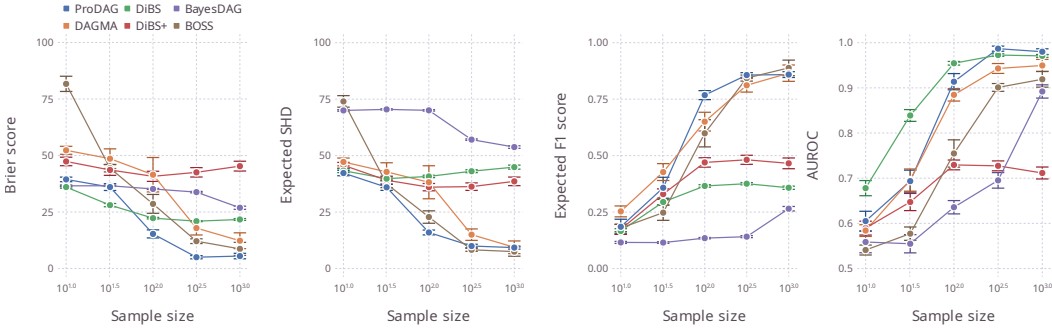

Figure 11: Performance on synthetic datasets generated from linear Erdős–Rényi DAGs with $p = 20$ nodes, $s = 40$ edges, and Gumbel noise. The averages (solid points) and standard errors (error bars) are measured over 10 independently and identically generated datasets.

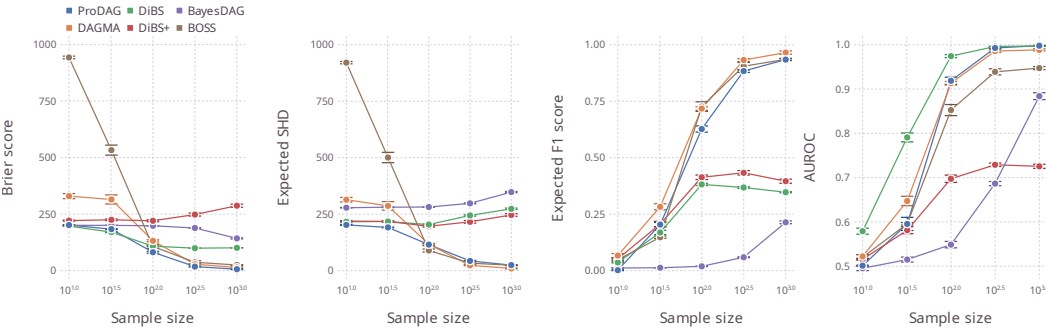

Figure 12: Performance on synthetic datasets generated from linear Erdős–Rényi DAGs with $p = 100$ nodes, $s = 200$ edges, and Gumbel noise. The averages (solid points) and standard errors (error bars) are measured over 10 independently and identically generated datasets.

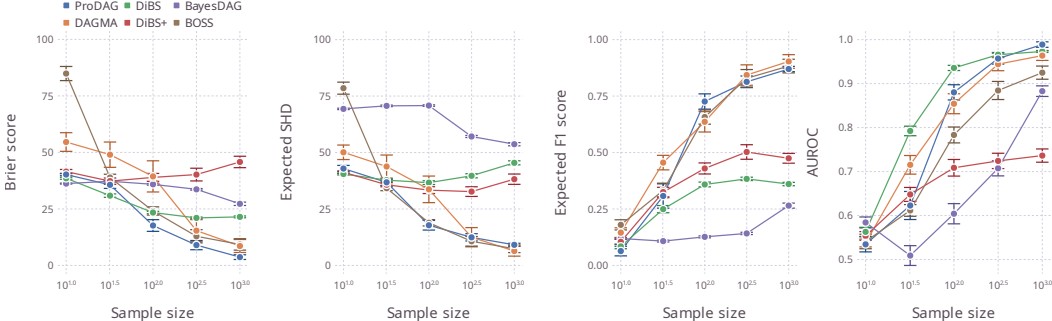

Figure 13: Performance on synthetic datasets generated from linear Erdős–Rényi DAGs with $p = 20$ nodes, $s = 40$ edges, and exponential noise. The averages (solid points) and standard errors (error bars) are measured over 10 independently and identically generated datasets.

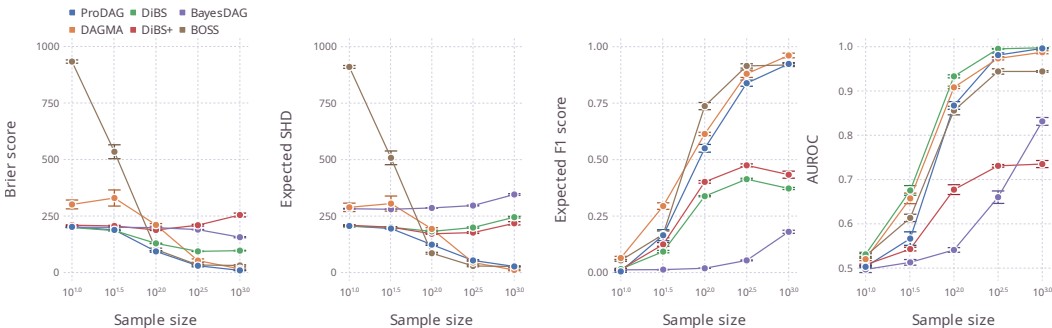

Figure 14: Performance on synthetic datasets generated from linear Erdős–Rényi DAGs with $p = 100$ nodes, $s = 200$ edges, and exponential noise. The averages (solid points) and standard errors (error bars) are measured over 10 independently and identically generated datasets.

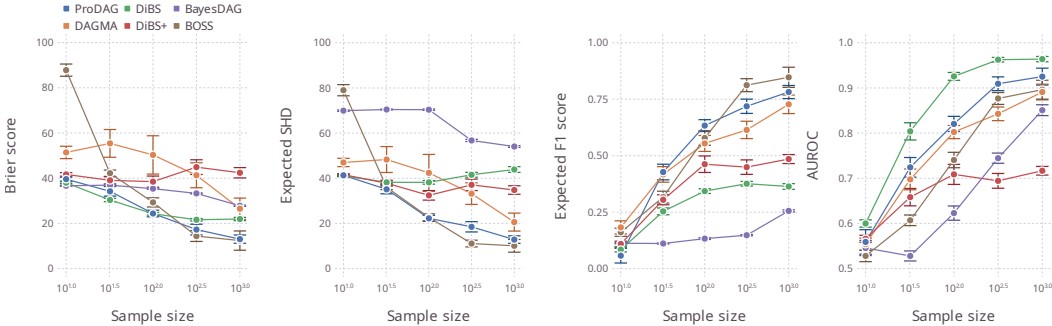

Figure 15: Performance on synthetic datasets generated from linear Erdős–Rényi DAGs with $p = 20$ nodes, $s = 40$ edges, and heteroscedastic Gaussian noise. The averages (solid points) and standard errors (error bars) are measured over 10 independently and identically generated datasets.

### G.4 Timings

We compare the run times of `ProDAG` with those of the Bayesian baselines in our main experiments. Table 4 reports timings on ($p = 20$ nodes, $s = 40$ edges) and nonlinear ($p = 10$ nodes, $s = 20$ edges) Erdős–Rényi DAGs with Gaussian noise. The timings show that `ProDAG` is faster than `BayesDAG`

Table 4: Run times in seconds with a sample size $n = 100$ for linear ($p = 20$ nodes) and nonlinear ($p = 10$ nodes) settings. The averages and standard errors are measured over 10 independently and identically generated datasets. The best value in each setting is indicated in bold.

|         | Linear         | Nonlinear       |
|---------|----------------|-----------------|
| ProDAG  | $30 \pm 7.0$   | $61 \pm 3.8$    |
| DiBS    | $\mathbf{8 \pm 0.7}$ | $\mathbf{40 \pm 0.4}$ |
| BayesDAG | $57 \pm 1.1$  | $122 \pm 5.5$   |

and slower than `DiBS`. However, `ProDAG`'s superior performance in uncertainty quantification and structure recovery makes it a strong choice, even with the higher computational cost over `DiBS`.

## H  Limitations

Though our approach has several favorable properties, it naturally has limitations, the main ones we highlight here. As with most Bayesian approaches that employ variational inference, ours does not guarantee that the learned variational posterior approximates the true posterior well. The possibility of a poor variational approximation is further complicated by the nonconvexity of the acyclicity constraint, giving rise to a nonconvex optimization problem that can be solved only to a stationary point. We remark, however, that nonconvexity is a property of most existing DAG learning methods.

## I  Societal impacts

DAG learning tools play a key role in causal inference, where the goal is to establish causal relationships between random variables. Our approach benefits from quantifying uncertainty in this process, which can help guide downstream decision-making. However, as with any DAG learning tool applied to observational data, one cannot causally interpret the DAGs learned by our approach without making empirically unverifiable assumptions such as causal sufficiency. Therefore, it is crucial to involve subject-matter experts capable of determining the validity of any such assumptions.

