# OpenReview forum: "ProDAG: Projected Variational Inference for Directed Acyclic Graphs"
_NeurIPS.cc/2025/Conference — NeurIPS 2025 poster_

### Official Review · Reviewer_7A4S · 2025-07-02

**Clarity:** 3
**Significance:** 3
**Originality:** 3
**Rating:** 4
**Confidence:** 4

**Summary:**

This paper proposes a novel Bayesian causal discovery method ProDAG for learning directed acyclic graphs (DAG) with uncertainty quantification. The prior on the adjacency matrix $W$ of the DAGs is induced by projecting $\tilde W\in\mathbb R^{p\times p}, \tilde W\sim\mathbb P$ onto its closest neighbor in the set of adjacency matrices. The author showed that the projection is unique and thus the prior is well-defined. Furthermore, the author introduced a variational inference method to learn the underlying DAG based on this prior. Experiments have shown that ProDAG performs competitively against the state-of-the-art (Bayesian) DAG learning methods.

**Questions:**

1. Do you assume knowledge of the true noise distribution for experiments on non-Gaussian data?
2. Is ProDAG robust if the log-likelihood function is misspecified? For example, using a Gaussian likelihood for data generated by non-Gaussian noise distributions, or an equal variance likelihood for a heterogeneous non-Gaussian noise distribution?
3. How sensitive are the results to the choice of $\mathbb P$?

**Ethical Concerns:**

["NO or VERY MINOR ethics concerns only"]

**Final Justification:**

The author has clarified the experimental setting and provided more extensive experiments which addressed most of my concerns. My remaining concern is whether the method is robust under heterogeneous noise distribution. However, given its solid methodology, I find it not a reason to reject the paper. That is, even if it is not the case (but the author should mention this), the paper still provide a solid variational inference method for causal discovery in several scenarios. Therefore, I raise my score to 4.

**Limitations:**

Yes

**Quality:**

3

**Strengths And Weaknesses:**

**Strengths**
1. The proposed method is novel and shows good performance for various noise distributions and graph structures.
2. The prior distribution is well-motivated and justified.
3. The projection procedure in the prior evaluation is efficient. Moreover, the author provides explicit gradient estimation for this projection.

**Weaknesses**
1. Identifiability. The author did not discuss what kind of DAGs the proposed model can identify. Since the VI procedure requires knowledge of the likelihood, it seems that the noise distribution has to be known to guarantee identifiability, which is a limitation that needs to be declared.
2. Some experiments are not sufficient. For example, the experiment for nonlinear data is limited to $d=10$, and those for non-Gaussian noise distributions are limited to $d=20$. Experiments on higher dimensions, e.g., $d=100$ would be helpful.
3. Time comparison with other Bayesian methods is not provided.

I will be happy to increase the score if my concerns can be addressed.

---

> ### Author Rebuttal · Authors · 2025-07-31
>
> We are grateful for your engagement and the constructive feedback on our submission.
>
> ## Weaknesses
>
> > Identifiability. The author did not discuss what kind of DAGs the proposed model can identify. Since the VI procedure requires knowledge of the likelihood, it seems that the noise distribution has to be known to guarantee identifiability, which is a limitation that needs to be declared.
>
> Thank you for highlighting this point. Indeed, identifiability depends on both the data-generating SEM (including the noise distribution) and the user-specified likelihood. For instance:
>
> - If the likelihood is correctly specified (e.g., Gaussian likelihood applied to Gaussian noise), the true DAG (or its equivalence class) is theoretically identifiable. In our experiments (e.g., Figure 3), we observe this behavior as the posterior increasingly concentrates on the true structure as $n$ grows; and
> - If the likelihood is misspecified (e.g., Gaussian likelihood applied to non-Gaussian noise), full identifiability may no longer hold. In this scenario, `ProDAG` will reflect the resulting uncertainty in its posterior.
>
> Thus, `ProDAG` captures and reports the inherent uncertainty determined by the combination of the true SEM and the chosen likelihood. We will explicitly clarify this subtlety in the revised manuscript.
>
> > Some experiments are not sufficient. For example, the experiment for nonlinear data is limited to $d=10$, and those for non-Gaussian noise distributions are limited to $d=20$. Experiments on higher dimensions, e.g., $d=100$ would be helpful.
>
> We added the following experiments with more nodes $p$ ($d$ in your notation) to address your concerns:
>
> 1. Nonlinear data with $p=20$ (double the previous dimension) in **Table R3**;
> 2. Gumbel noise data with $p=100$ in **Table R4**; and
> 3. Exponential noise data with $p=100$ in **Table R5**.
>
> Due to NeurIPS image restrictions this year, we report a snapshot of the experiments at $n=100$.
>
> **Table R3** Larger nonlinear experiment with $p=20$.
>
> |            | Brier score | Exp. SHD   | Exp. F1 score (%) | AUROC (%) |
> |:-----------|:------------|:-----------|:------------------|:----------|
> | `ProDAG`   | 25 ± 1.5    | 49 ± 3.7   | 41 ± 2.1          | 81 ± 2.8  |
> | `DAGMA`    | 56 ± 8.1    | 55 ± 7.9   | 32 ± 3.9          | 62 ± 2.0  |
> | `BayesDAG` | 34 ± 0.3    | 67 ± 1.6   | 14 ± 0.5          | 67 ± 2.0  |
>
>
> **Table R4** Larger Gumbel noise experiment with $p=100$.
>
> |            | Brier score | Exp. SHD   | Exp. F1 score (%) | AUROC (%) |
> |:-----------|:------------|:-----------|:------------------|:----------|
> | `ProDAG`   | 81 ± 4.3    | 115 ± 4.6  | 63 ± 1.4          | 92 ± 0.8  |
> | `DAGMA`    | 131 ± 4.0   | 116 ± 3.3  | 72 ± 0.8          | 91 ± 0.4  |
> | `DiBS`     | 108 ± 1.6   | 204 ± 2.0  | 38 ± 0.5          | 97 ± 0.3  |
> | `DiBS+`    | 221 ± 4.2   | 196 ± 3.3  | 41 ± 1.0          | 70 ± 0.8  |
> | `BayesDAG` | 198 ± 0.2   | 282 ± 0.9  | 2 ± 0.1           | 55 ± 0.7  |
>
> **Table R5** Larger exponential noise experiment with $p=100$.
>
> |            | Brier score | Exp. SHD   | Exp. F1 score (%) | AUROC (%) |
> |:-----------|:------------|:-----------|:------------------|:----------|
> | `ProDAG`   | 93 ± 3.4    | 123 ± 3.2  | 55 ± 1.7          | 87 ± 0.9  |
> | `DAGMA`    | 215 ± 6.6   | 197 ± 6.1  | 61 ± 0.8          | 91 ± 0.3  |
> | `DiBS`     | 129 ± 0.8   | 182 ± 0.9  | 34 ± 0.3          | 93 ± 0.3  |
> | `DiBS+`    | 188 ± 2.4   | 172 ± 2.1  | 40 ± 0.6          | 68 ± 1.1  |
> | `BayesDAG` | 199 ± 0.1   | 286 ± 1.2  | 2 ± 0.0           | 54 ± 0.5  |
>
> The results align with our earlier findings, confirming that `ProDAG` scales to larger, more complex settings. Note that `DiBS`/`DiBS+` ran out of memory in the big nonlinear experiment so are not included. Likewise, `BOSS` is not reported in Table R3 because it does not accommodate nonlinear models, and Tables R4 and R5 because it encountered errors.
>
> > Time comparison with other Bayesian methods is not provided.
>
> We added a time comparison in **Table R6** for linear and nonlinear DAGs (corresponding to the experiments in Figures 3 and 5 of the paper). These timings show that `ProDAG` is faster than `BayesDAG` and slower than `DiBS`. However, `ProDAG`'s superior performance in accuracy and uncertainty quantification makes it a strong choice, even with the higher computational cost compared to `DiBS`.
>
> **Table R6** Run times (seconds) against Bayesian baselines.
>
> |              | Linear | Nonlinear |
> |:-------------|:-------|:----------|
> | `ProDAG`     | 30 ± 7 | 61 ± 4    |
> | `DiBS/DiBS+` | 8 ± 0  | 40 ± 0    |
> | `BayesDAG`   | 57 ± 1 | 122 ± 5   |
>
> ## Questions
>
> > Do you assume knowledge of the true noise distribution for experiments on non-Gaussian data?
>
> No, the model is always fit with a Gaussian likelihood $x=W^\top x+\epsilon,\epsilon\sim N(0,\Sigma)$ independent of the true noise distribution in the simulated data.
>
> > Is ProDAG robust if the log-likelihood function is misspecified? For example, using a Gaussian likelihood for data generated by non-Gaussian noise distributions, or an equal variance likelihood for a heterogeneous non-Gaussian noise distribution?
>
> Yes, as in our non-Gaussian experiments, when we deliberately apply the Gaussian likelihood to data generated with Gumbel or exponential noise, `ProDAG`'s performance is roughly similar to the Gaussian noise case, indicating it is resilient to moderate likelihood misspecification.
>
> > How sensitive are the results to the choice of $\mathbb{P}$?
>
> Any continuous prior on $\tilde{W}$ is pushed through the same projection $\tilde{W}\mapsto W$, so the induced prior is largely governed by the $\ell_1$ ball and acyclicity constraints, not by the exact density of $\mathbb{P}$. We therefore use an isotropic Gaussian, which is quite general. Having said that, some priors such as Laplace, which shrink towards zero, are not well-suited as they can combine with the $\ell_1$-ball constraint to induce double shrinkage.

---

> ### Comment · Reviewer_7A4S · 2025-08-01
> **Reply to the rebuttal**
>
> I thank the author for the detailed reply, which addressed most of my concerns. Thus, I will raise my score to 4. However, I still have a follow-up question on the noise variance. Do you assume equal noise variance in the experiments? If that is the case, I would recommend that the author include experiments on column-standardized data [1] or varying variance, as recent works [1,2,3] have identified limitations of differentiable causal discovery methods on data with unequal variance. Clarifying this would further strengthen the paper.
>
> [1] Reisach, A., Seiler, C., & Weichwald, S. (2021). Beware of the simulated dag! causal discovery benchmarks may be easy to game. Advances in Neural Information Processing Systems, 34, 27772-27784.
>
> [2] Ng, I., Huang, B., & Zhang, K. (2024). Structure learning with continuous optimization: A sober look and beyond. In Causal Learning and Reasoning (pp. 71-105). PMLR.
>
> [3] Berrevoets, J., Raymaekers, J., Van der Schaar, M., Verdonck, T., & Yao, R. (2025). Differentiable Causal Structure Learning with Identifiability by NOTIME. In International Conference on Artificial Intelligence and Statistics (pp. 3115-3123). PMLR.

---

> > ### Author Response · Authors · 2025-08-04
> >
> > Thank you for considering our response and for your constructive follow-up. The current experiments use homoscedastic noise (i.e., equal variance) across nodes. We appreciate that [1–3] highlight unequal variance as a potential limitation of differentiable DAG learning methods, and we commit to including further experiments with heteroscedastic noise (or column-standardized data) in the final version.

---

### Official Review · Reviewer_agaZ · 2025-07-03

**Clarity:** 3
**Significance:** 3
**Originality:** 3
**Rating:** 5
**Confidence:** 3

**Summary:**

This paper introduces a method for learning DAGs that addresses the challenging problem of uncertainty quantification in causal structure discovery. The main innovation is the development of "projected distributions" that have support directly on the space of sparse DAGs by projecting samples from continuous distributions onto the set of acyclic matrices using a combination of acyclicity and sparsity constraints. The authors prove these projected distributions are mathematically valid, develop efficient algorithms for the required projections, and demonstrate through a range of experiments that their method performs well in both accuracy and uncertainty quantification.

**Questions:**

1. Can you demonstrate the method on larger networks (p=200-500), or propose any algorithmic improvements that could
significantly increase the scaleability of the approach?

2. Can you determine which components are responsible for the performance improvements? Some ablations could help with this, such as  comparing against a simpler baselines with the same acyclicity constraints without projected distributions.

3. Can you provide recommendations about the settings in which your approach should be chosen over existing methods, given the modest improvements and higher computational cost?

4. Can you provide more analyses as to when the variational approximation works well? This could include diagnostics or an experiment comparing against exact posteriors on a small synthetic problem.

**Ethical Concerns:**

["NO or VERY MINOR ethics concerns only"]

**Limitations:**

The limitations are discussed above. I do not see any potential negative societal impact of the work.

**Quality:**

4

**Strengths And Weaknesses:**

This is a strong paper that makes solid theoretical and practical contributions to Bayesian DAG learning. The core idea of projected distributions addresses an important problem, as most existing Bayesian DAG methods either do not guarantee exact acyclicity or require complex discrete inference procedures. The method appears technically sound and shows improvements over baselines, though gains are often small. The projected distribution idea could likely be applied beyond DAGs to other constrained structure learning problems, increasing the significance of the work.

Strengths:

Quality: The technical execution is very good, with rigorous mathematical foundations and proofs, and comprehensive experimental evaluation in both linear and nonlinear SEMs with multiple metrics and baselines. The variational inference framework is well-designed and the extension to nonlinear DAGs via neural networks is natural, though the method still inherits standard limitations like its cubic complexity and approximate posteriors from variational inference.

Clarity: The paper is well-written, with the projected distribution concept presented intuitively and the theoretical framework clearly explained. The novel ideas are integrated nicely with known results.

Significance: This addresses an important problem in causal inference with practical implications for uncertainty quantification in downstream tasks. The method shows strong uncertainty quantification compared to frequentist approaches and performs well for large sample sizes where other Bayesian methods fall off, with particularly good results on the real protein signaling dataset.

Originality: The key innovation is to define a novel distribution P(W) over the space of DAG adjacency matrices, which can be used in a variational framework. The approach seems innovative compared to previous approaches, especially this contribution of defining this distribution with continuous support. The entire framework also combines the new ideas with known results/tools nicely.

Weaknesses:

Quality: The method inherits standard limitations including cubic complexity, which limits its scalability (experiments only go to p=100). The use of variational inference does not guarantee good approximation to the true posterior. The non-convexity of the acyclicity constraint means optimization can only reach stationary points rather than global optima.

Clarity: While generally well-written, some important experimental details are deferred to appendices rather than being in the main text, and the combination of multiple existing techniques (though novel in combination) makes it hard to work out which components drive the performance improvements.

Significance: The performance improvements over baselines are consistent but small, and the scalability limitations to networks with around 100 variables may limit practical applicability to larger real-world problems where the method's benefits would be most valuable.

Originality: While the overall framework is novel, the individual technical components such as the acyclicity constraints, L1-ball projections,  and variational inference) are all existing techniques, so the contribution is primarily in their creative combination rather than fundamental algorithmic innovations. Nonetheless there are some significant innovations in the work.

---

> ### Author Rebuttal · Authors · 2025-07-31
>
> Thank you for taking the time to engage with our work and for your thoughtful feedback.
>
> ## Weaknesses
>
> > The method inherits standard limitations including cubic complexity, which limits its scalability (experiments only go to p=100). The use of variational inference does not guarantee good approximation to the true posterior. The non-convexity of the acyclicity constraint means optimization can only reach stationary points rather than global optima.
>
> It is correct that non-convexity means we can only attain a local optimum, though this property is true of all differentiable DAG learning methods (e.g., `DAGMA`). Regarding scalability, we now include larger experiments (see our response below).
>
> > While generally well-written, some important experimental details are deferred to appendices rather than being in the main text, and the combination of multiple existing techniques (though novel in combination) makes it hard to work out which components drive the performance improvements.
>
> With the extra page allowed in the camera ready version, we will move additional experimental details (e.g., data generation and hyperparameter tuning) from the appendix to the main text. Regarding the source of performance gains, please see our response below concerning component attribution and ablations.
>
> > The performance improvements over baselines are consistent but small, and the scalability limitations to networks with around 100 variables may limit practical applicability to larger real-world problems where the method's benefits would be most valuable.
>
> We agree that practical applicability depends on scalability. As noted in our response below, we now include results on the Andes network with $p=223$ in **Table R2** (see response to Reviewer VF9w), which demonstrate that `ProDAG` scales beyond $p=100$ while maintaining its performance advantages.
>
> > While the overall framework is novel, the individual technical components such as the acyclicity constraints, L1-ball projections, and variational inference) are all existing techniques, so the contribution is primarily in their creative combination rather than fundamental algorithmic innovations. Nonetheless there are some significant innovations in the work.
>
> The individual tools (acylclicity characterizations, $\ell_1$ balls) are indeed known, but their composition to define a projected prior is, as you say, new. No prior work samples exact sparse DAGs by projecting continuous draws and then instantiates that construction in a variational framework. This step is the key innovation that drives `ProDAG`'s gains.
>
> ## Questions
>
> > Can you demonstrate the method on larger networks (p=200-500), or propose any algorithmic improvements that could significantly increase the scaleability of the approach?
>
> We now include experiments on the $p=223$ Andes network from bnlearn (**Table R2**). On this larger, real‑world‑structured benchmark, `ProDAG` continues to delver excellent performance and represents one of the largest Bayesian DAG evaluations reported to date.
>
> > Can you determine which components are responsible for the performance improvements? Some ablations could help with this, such as comparing against a simpler baselines with the same acyclicity constraints without projected distributions.
>
> We believe that `DAGMA`, a key baseline in our experiments, reflects the ablation you propose: it enforces the same acyclicity constraints but omits projected distributions. `DAGMA` can also be interpreted as a MAP version of `ProDAG`. The consistent improvements over `DAGMA` underscore the value of modeling full posteriors directly over DAGs, made possible by our projected variational framework. In practice, these gains trace to two components: (i) projected DAG distributions that place mass on sparse, exactly acyclic structures, and (ii) variational inference that leverages this distributional support to propagate uncertainty during learning.
>
> > Can you provide recommendations about the settings in which your approach should be chosen over existing methods, given the modest improvements and higher computational cost?
>
> Choose `ProDAG` when one of these situations arise:
>
> 1. You need well-calibrated posterior uncertainty (e.g., decision making, active experimentation);
> 2. The number of observations is limited, so precise sparsity and exact zeros aid recovery; or
> 3. Exact acyclicity and hard sparsity matter for interpretability (e.g., policy, biology).
>
> > Can you provide more analyses as to when the variational approximation works well? This could include diagnostics or an experiment comparing against exact posteriors on a small synthetic problem.
>
> While exact posterior comparisons via MCMC are nontrivial, we report Brier score, SHD, F1 score, and AUROC as direct assessments of the variational approximation’s structure learning capacity. To further stress-test this approximation, we added new higher-dimensional experiments on nonlinear and non-Gaussian data (**Tables R3–R5**), and the Andes network. In all cases, `ProDAG` maintains strong performance, demonstrating robustness of the approximation across a range of challenging settings.

---

### Official Review · Reviewer_VF9w · 2025-07-07

**Clarity:** 2
**Significance:** 3
**Originality:** 3
**Rating:** 4
**Confidence:** 2

**Summary:**

This paper utilize Bayesian variational inference framework to construct a valid distribution on the space of sparse DAGs for the causal discovery task. The proposed method ProDAG models the prior and variational posterior of potential DAGs. For this purpose the authors use the continuous reformulations of acyclicity constraints and perform gradient descent based on that.

**Questions:**

Below I provide my questions.

* How do the author set the value for $\lambda$.
* The authors said that the projection (2) is unique. Why can there not be two different $\tilde{W}$ with equal distance from $W$?
* The authors mentioned that step 1 projects the matrix $\tilde{W}$ onto the set of acyclic matrices through equation 4. How does the algorithm have access to $W$ to calculate the subtraction here?

**Ethical Concerns:**

["NO or VERY MINOR ethics concerns only"]

**Final Justification:**

This work has some novel contributions and the authors also added new experiments according to my suggestion. Thus, I keep my positive score.

**Limitations:**

yes.

**Quality:**

3

**Strengths And Weaknesses:**

Below I provide my comments.

# Strengths
Figure 1 is very helpful to understand the algorithm concepts. The idea of using variational inference for causal discovery appears novel.

# Weaknesses

* The following works also considers a continuous constraint optimization problem
The authors should compare their methods to the following works qualitatively and use some of them as benchmarks.

1. Brouillard, Philippe, et al. "Differentiable causal discovery from interventional data." Advances in Neural Information Processing Systems 33 (2020): 21865-21877.
2. Bhattacharya, Rohit, et al. "Differentiable causal discovery under unmeasured confounding." International Conference on Artificial Intelligence and Statistics. PMLR, 2021.
3. Faria, Gonçalo Rui Alves, Andre Martins, and Mário AT Figueiredo. "Differentiable causal discovery under latent interventions." Conference on Causal Learning and Reasoning. PMLR, 2022.
4. He, Yue, et al. "Daring: Differentiable causal discovery with residual independence." Proceedings of the 27th ACM SIGKDD conference on knowledge discovery & data mining. 2021.
5. Lopez, Romain, et al. "Large-scale differentiable causal discovery of factor graphs." Advances in Neural Information Processing Systems 35 (2022): 19290-19303.

* The authors should provide more details on the “path-following algorithm”.
* The authors should provide some intuition on “implicit function theorem.” and “Karush-Kuhn- Tucker (KKT) conditions of the projection” in the main paper.
* The role of $l_1$ ball is not clear. The author should discuss how large or small $l_1$ ball affects the algorithm performance. Also it is not clear how $l_1$ is chosen.
* It is not clear how the authors obtain $q_{\theta}(\tilde{W})$ with variational inference in their algorithm. I would suggest the authors discussing this in detail.
* The authors should show more experimental results with real-world data. They might check https://www.bnlearn.com/bnrepository/ for this purpose.
* Figure 2 might be replaced with a table and the space can be used for describing other concepts in detail.

---

> ### Author Rebuttal · Authors · 2025-07-31
>
> Thank you for your thoughtful review and engagement with the paper.
>
> ## Weaknesses
>
> > The following works also considers a continuous constraint optimization problem The authors should compare their methods to the following works qualitatively and use some of them as benchmarks.
>
> Thank you for these references. We agree they relate to our work but target different settings and all are non-Bayesian. Brouillard et al. (2020), Faria et al. (2022) and Lopez et al. (2022) rely on interventional data, whereas `ProDAG` assumes purely observational data. Bhattacharya et al. (2021) also uses observational data but learns acyclic directed mixed graphs (graphs with bidirected edges), not DAGs. He et al. (2021) adds a residual‑independence regularizer to frequentist `NOTEARS`‑type models, but it is unclear how to embed this within `ProDAG`’s Bayesian projection prior. We will cite the papers and discuss the extension of `ProDAG` to interventional data as future work.
>
> > The authors should provide more details on the “path-following algorithm”.
>
> We agree that the path-following algorithm (presented in Appendix B) would benefit from additional explanatory text to complement the algorithmic details. We will revise the appendix to include a more descriptive explanation of the algorithm’s steps.
>
> > The authors should provide some intuition on “implicit function theorem.” and “Karush-Kuhn- Tucker (KKT) conditions of the projection” in the main paper.
>
> We agree. While the full details were placed in Appendix C due to space limitations, we will include descriptions in the main text in the final version.
>
> > The role of $l_1$ ball is not clear. The author should discuss how large or small $l_1$ ball affects the algorithm performance. Also it is not clear how $l_1$ is chosen.
>
> The $\ell_1$ ball constraint is included to sample sparse graphs; a larger $\ell_1$ ball mean more edges while a smaller $\ell_1$ ball means fewer edges. In our numerical experience, the algorithm's speed is not particularly sensitive to the size of the ball, though smaller balls tend to converge slightly faster than larger balls. The size $\lambda$ of the ball is chosen using a validation set (see Appendix E).
>
> > It is not clear how the authors obtain $q_\theta(\tilde{W})$ with variational inference in their algorithm. I would suggest the authors discussing this in detail.
>
> The distribution $q_\theta(\tilde{W})$ is the variational family chosen by the user (see the last paragraph of Section 5.1) such as a multivariate Gaussian, and its parameters $\theta$ are optimized via gradient descent to maximize the ELBO. The projected distribution over DAGs, $q_\theta(W)$, is implicitly defined by applying the projection to samples from $q_\theta(\tilde{W})$.
>
> > The authors should show more experimental results with real-world data. They might check bnrepository for this purpose.
>
> Thank you for the reference to this great repository. We have now added **Table R2**, which reports results on the Andes network ($p=223$) from the repository, drawing $n=100$ samples from its DAG with our usual data generation pipeline. This large, real‑world‑structured benchmark shows that `ProDAG` performs strongly on very high-dimensional DAGs, achieving the lowest Brier score and SHD. While `DAGMA` scores well on F1 and AUROC, `ProDAG` provides superior calibration and structure recovery.
>
> **Table R2** Experiments on real-world Andes DAG with $p=223$ nodes and $s=338$ edges. We report 8 runs here due to time constraints and will extend to 10 in the final version, but results are already clearly differentiated.
>
> |            | Brier score | Exp. SHD   | Exp. F1 score (%) | AUROC (%) |
> |:-----------|:------------|:-----------|:------------------|:----------|
> | `ProDAG`   | 143 ± 3.4   | 198 ± 4.7  | 39 ± 2.0          | 87 ± 0.8  |
> | `DAGMA`    | 337 ± 78.7  | 306 ± 77.8 | 65 ± 4.9          | 89 ± 0.6  |
>
> To our knowledge, this experiment represents one of the largest Bayesian DAG evaluations reported to date. Aside from `ProDAG`, `DAGMA` was the only method to successfully run on this large dataset. The other Bayesian baselines encountered difficulties: `DiBS` and `DiBS+` produced nonsensical results (Brier scores and SHDs in the tens of thousands), indicative of convergence issues, while `BayesDAG` ran out of memory (24 GB available). `BOSS` encountered crashes.
>
> > Figure 2 might be replaced with a table and the space can be used for describing other concepts in detail.
>
> If space is a concern in the final version, we will move Figure 2 to the appendix to make room for additional explanations in the main text.
>
> ## Questions
>
> > How do the author set the value for $\lambda$.
>
> The parameter $\lambda$ is chosen by maximizing the expected likelihood on a validation set. See Appendix E for details.
>
> > The authors said that the projection (2) is unique. Why can there not be two different $\tilde{W}$ with equal distance from $W$?
>
> Uniqueness means that for almost every draw of $\tilde{W}$ the projection (2) has one minimizer, so $\operatorname{pro}_\lambda(\tilde{W})$ is single‑valued for each fixed $\tilde{W}$. This result does not imply injectivity: different $\tilde{W}$ can still project to the same $W$, but no single $\tilde{W}$ has more than one optimal projection. We will clarify this point in the text.
>
> > The authors mentioned that step 1 projects the matrix $\tilde{W}$ onto the set of acyclic matrices through equation 4. How does the algorithm have access to $W$ to calculate the subtraction here?
>
> The $W$ in (4) appearing inside $f_\mu(W;\tilde{W})=\frac{\mu}{2}\|\tilde{W}-W\|_F^2+h(W)$ is the current optimization variable. The optimizer repeatedly updates this variable while it computes the difference $\tilde{W}-W$ until $f$ is minimized. After convergence we reuse the same symbol to denote the minimizer, so the duplicate notation is purely stylistic and we never assume prior knowledge of the solution.

---

### Official Review · Reviewer_e12N · 2025-07-22

**Clarity:** 4
**Significance:** 3
**Originality:** 3
**Rating:** 5
**Confidence:** 4

**Summary:**

This paper introduces ProDAG, a Bayesian structure learning method that employs variational inference to infer an approximate posterior over DAGs given observational data. Unlike prior variational approaches which either (i) enforce acyclicity in the variational family or (ii) modify the prior to penalize cyclicity, ProDAG takes the novel approach of choosing an unconstrained variational family that induces a distribution over DAGs through the use of a many-to-one projection operation. It is shown that this defines a valid distribution over DAGs (weight matrices), and how to propagate gradients through the projection operation. Empirical results show that ProDAG outperforms existing variational Bayesian structure learning methods on standard metrics, particularly as the sample size increases.

**Questions:**

- Intuitively, why is it possible to derive the gradients of the optimal solution (Apx C) in closed form but not the optimal solution itself? Given that the iterative procedure only produces an approximation solution to the projection, is there a significant discrepancy between the exact autograd gradients for the approximate solution and the gradients of the optimal solution?
- Why did the authors choose a multivariate Gaussian variational posterior, given that the true posteriors can be multimodal? (unless the projection step induces multimodality). It would be interesting to see some examples in low dimensions to visualize the effect of the projection step.
- Did the authors try learning a variational posterior with a Laplace-style prior instead of the hard sparsity projection? If so and it performed worse, are there some qualitative insights into why the hard projection step should be preferred?
- How did the authors tune the sparsity hyperparameter $\lambda$ on the validation data? How sensitive is the performance of the method to the choice of $\lambda$ (i.e. would it be reasonable to simply choose a single $\lambda$ value and use it across all experiments)?

**Ethical Concerns:**

["NO or VERY MINOR ethics concerns only"]

**Final Justification:**

The authors have fully addressed my concerns regarding experimental evaluation, and I support acceptance of the paper.

**Limitations:**

Yes, limitations are addressed in the appendix.

**Quality:**

3

**Strengths And Weaknesses:**

Overall, this is a high quality paper introducing an interesting approach for Bayesian structure learning. The hard projection step onto weight matrices with a particular sparsity is a novel idea and is shown to lead to strong empirical performance. The idea is backed up with solid theoretical work showing the measurability of the projection mapping and derivation of the ELBO and gradients. The clarity of the paper is also excellent with the key ideas being described well in the main paper with full details presented in the Appendix.

Regarding weaknesses, while the empirical evaluation shows strong performance, I would like to have seen some more insight into *why* ProDAG performs better than existing approaches (see questions below). Additionally, the empirical evaluation compares to variational approaches such as BayesDAG and DiBS that employ augmented spaces. However, for completeness there should also be comparison to other types of Bayesian methods, in particular (i) permutation-based variational methods such as BCDNets; and (ii) MCMC-based approaches such as Gadget [1] which may exhibit qualitatively different behavior (e.g. consistency of MCMC). The related work section should also mention some of these MCMC-based approaches. Finally, a seeming limitation is that ProDAG requires the sparsity hyperparameter $\lambda$ to be set in advance, and due to the hard projection constraint this may inevitably lead to errors even in the large-data limit (i.e. if one sets $\lambda < d(d-1)/2$ then one could never learn the complete graph even if it is the true graph).


[1] Towards Scalable Bayesian Learning of Causal DAGs. Viinikka et al. (2020). NeurIPS

---

> ### Author Rebuttal · Authors · 2025-07-31
>
> We appreciate your time and attention in reviewing our work.
>
> ## Weaknesses
>
> > Regarding weaknesses, while the empirical evaluation shows strong performance, I would like to have seen some more insight into why ProDAG performs better than existing approaches (see questions below).
>
> `ProDAG` places both the prior and variational posterior directly on DAG space, enforcing exact acyclicity through projection. Competing methods (i) operate in relaxed spaces with soft penalties that cannot guarantee acyclicity, or (ii) introduce discrete variables (e.g., permutations) that complicate optimization. See the discussion in Section 2. By pushing irrelevant edges to exact zeros while keeping the ELBO differentiable, `ProDAG` yields sharper uncertainty estimates and lower SHD scores.
>
> > Additionally, the empirical evaluation compares to variational approaches such as BayesDAG and DiBS that employ augmented spaces. However, for completeness there should also be comparison to other types of Bayesian methods, in particular (i) permutation-based variational methods such as BCDNets; and (ii) MCMC-based approaches such as Gadget [1] which may exhibit qualitatively different behavior (e.g. consistency of MCMC). The related work section should also mention some of these MCMC-based approaches.
>
> For permutation-related variational methods such as `BCDNets`, we note that we have, in fact, compared to `BayesDAG` which is representative of the state-of-the-art in this class of methods and has been shown in its original paper to exceed `BCDNets`'s performance. See Section 3.2 of the `BayesDAG` paper for the equivalence to permutation approaches and Section 6 for the comparisons against `BCDNets`. For MCMC-based methods, we have now added comparisons against `Gadget` with $p=20$ and $n=100$. **Table R1** shows it substantially underperforms `ProDAG`. This underperformance is not due to `Gadget`’s configuration, as we evaluated many settings and report the best-performing one. We will expand the related work section to include a discussion of `Gadget` and MCMC-based methods.
>
> **Table R1** Comparisons against `Gadget`.
>
> |          | Brier score | Exp. SHD | Exp. F1 score (%) | AUROC (%) |
> |:---------|:------------|:---------|:------------------|:----------|
> | `ProDAG` | 13 ± 1.8    | 14 ± 1.5 | 78 ± 2.7          | 91 ± 1.6  |
> | `Gadget` | 52 ± 1.8    | 37 ± 0.5 | 17 ± 1.5          | 65 ± 0.5  |
>
> > Finally, a seeming limitation is that ProDAG requires the sparsity hyperparameter $\lambda$ to be set in advance, and due to the hard projection constraint this may inevitably lead to errors even in the large-data limit (i.e. if one sets $\lambda<d(d-1)/2$ then one could never learn the complete graph even if it is the true graph).
>
> We treat $\lambda$ purely as a regularisation knob and apply a grid search, so the tuning procedure (see our response below) selects the $\lambda$ best supported by the data. The grid search includes $\lambda=\infty$ so the true graph, however dense, is never ruled out. Only an intentionally small $\lambda$ would exclude it.
>
> ## Questions
>
> > Intuitively, why is it possible to derive the gradients of the optimal solution (Apx C) in closed form but not the optimal solution itself? Given that the iterative procedure only produces an approximation solution to the projection, is there a significant discrepancy between the exact autograd gradients for the approximate solution and the gradients of the optimal solution?
>
> The projection has no closed-form solution because it solves a high-dimensional non-convex problem, but once a stationary point $W^\star$ is attained its KKT conditions are algebraic relations that can be implicitly differentiated, giving the compact gradient formulas in Proposition 2 (in short, local sensitives are easier to obtain than global minimizers). The gradient formulas are evaluated at the $W^\star$ returned by the projection, so they exactly coincide with the gradient autograd would produce at the same point, i.e., there is no discrepancy.
>
> > Why did the authors choose a multivariate Gaussian variational posterior, given that the true posteriors can be multimodal? (unless the projection step induces multimodality). It would be interesting to see some examples in low dimensions to visualize the effect of the projection step.
>
> We choose a Gaussian in the unconstrained space $\tilde{W}$ for differentiable reparameterization, closed-form KL, and efficient optimization. We stress, however, that the projection $\tilde{W}\mapsto W$ does indeed induce a non-Gaussian, *multimodal posterior* over DAGs. We will clarify this point explicitly in the revised text. We are not allowed to include visualizations in the rebuttal, but Figure 1 in the paper roughly illustrates the effect of the projection.
>
> > Did the authors try learning a variational posterior with a Laplace-style prior instead of the hard sparsity projection? If so and it performed worse, are there some qualitative insights into why the hard projection step should be preferred?
>
> Yes, during prototyping we evaluated the DAG projection under a Laplace prior and saw noticeably higher SHD scores because Laplace only shrinks edges toward zero and does not set them exactly to zero, so the resulting graphs were too dense. In contrast, the $\ell_1$ ball prior assigns positive probabilities on exact zeros and thus yields crisper sparsity patterns and better edge recovery. See Xu & Duan (2023, arXiv:2006.01340) for a detailed comparison of $\ell_1$ ball versus Laplace.
>
> > How did the authors tune the sparsity hyperparameter $\lambda$ on the validation data? How sensitive is the performance of the method to the choice of $\lambda$ (i.e. would it be reasonable to simply choose a single $\lambda$ value and use it across all experiments)?
>
> We sweep $\lambda$ over ten values from $\lambda_\min=0$ to $\lambda_\max$ ($\lambda_\max$ is the average $\ell_1$-norm of $W$ from the learned posterior when $\lambda=\infty$) and pick the value that maximizes the expected likelihood on a held-out 10% validation set (see Appendix E). Because $\lambda$ directly controls graph sparsity, performance drops if it is set too low (underconnected) or too high (overconnected), so exploring a few points on a grid is preferable to a single value. We apply the same grid‑search to every method with a sparsity parameter.

---

> > ### Comment · Reviewer_e12N · 2025-08-09
> >
> > Thanks for the detailed response, particularly clarifying that the hyperparameter search is based on validation expected log-likelihood which addresses my concern about the sparsity. I will update my final review accordingly.

---

> > > ### Author Response · Authors · 2025-08-09
> > >
> > > Many thanks for considering our rebuttal. We’re glad your concerns regarding sparsity have been addressed and appreciate your updated assessment.

---

### Comment · Area_Chair_PcMb · 2025-08-08

We only have a short time left for the discussion period, and many reviewers have not yet responded. If you have further comments, I'm sure the authors would appreciate the chance for more discussion while they are able to respond!

---

### Note · Authors · 2025-08-12

We thank the reviewers for their constructive feedback and engagement. We believe we have addressed the major concerns in the rebuttal and discussion, with several reviewers confirming resolution of their points and, in some cases, raising their scores. As part of the rebuttal, we added larger-scale experiments, included MCMC baselines, and conducted robustness checks under non-Gaussian noise. These additions indicate that `ProDAG` scales, performs well against strong alternatives, and provides well-calibrated uncertainty. We also clarified $\lambda$ tuning, identifiability, and related work. Overall, reviewers acknowledged the novelty, theoretical rigor, and practical value of combining exact DAG projection with variational inference. We hope the additional results and clarifications convey `ProDAG`’s potential as a useful and principled contribution to Bayesian DAG learning.

---

### Decision · Program_Chairs · 2025-09-17

**Decision:**

Accept (poster)

**Comment:**

This paper addresses the problem of learning a distribution over directed acyclic graphs by reformulating the discrete posterior over graphs into a relaxed distribution with a projection operator, which is in turn approximated with a variational posterior. Reviewers agreed this work is of high quality and makes an important contribution. I ask the authors to make sure to honor their commitment to include further experiments with heteroscedastic noise in the final version.